# Statistical and Causal Robustness for Causal Null Hypothesis Tests

**Junhui Yang**[1,*]     **Rohit Bhattacharya**[2,*]     **Youjin Lee**[3]     **Ted Westling**[1]

[1]Department of Mathematics and Statistics, University of Massachusetts Amherst, Amherst, Massachusetts, USA
[2]Department of Computer Science, Williams College, Williamstown, Massachusetts, USA
[3]Department of Biostatistics, Brown University, Providence, Rhode Island, USA
[*]Equal contributors

## Abstract

Prior work applying semiparametric theory to causal inference has primarily focused on deriving estimators that exhibit statistical robustness under a prespecified causal model that permits identification of a desired causal parameter. However, a fundamental challenge is correct specification of such a model, which usually involves making untestable assumptions. Evidence factors is an approach to combining hypothesis tests of a common causal null hypothesis under two or more candidate causal models. Under certain conditions, this yields a test that is valid if at least one of the underlying models is correct, which is a form of causal robustness. We propose a method of combining semiparametric theory with evidence factors. We develop a causal null hypothesis test based on joint asymptotic normality of $K$ asymptotically linear semiparametric estimators, where each estimator is based on a distinct identifying functional derived from each of $K$ candidate causal models. We show that this test provides both statistical and causal robustness in the sense that it is valid if at least one of the $K$ proposed causal models is correct, while also allowing for slower than parametric rates of convergence in estimating nuisance functions. We demonstrate the effectiveness of our method via simulations and applications to the Framingham Heart Study and Wisconsin Longitudinal Study.

## 1 INTRODUCTION

Prior work at the intersection of semiparametric theory and causal inference has primarily focused on deriving estimators that possess statistical robustness properties under a prespecified causal model that permits identification of a causal parameter of interest. For example, in the backdoor causal model where treatment assignment is assumed to be ignorable given observed covariates, the average causal effect (ACE) is identified via the backdoor formula [Robins, 1986, Pearl, 2009], and the augmented inverse probability weighted estimator (AIPW) of this parameter [Bang and Robins, 2005] exhibits statistical robustness to specification of the propensity score and outcome regression estimators. In particular, the AIPW estimator is *doubly robust*, meaning that it is consistent if either the propensity score or outcome regression estimator is consistent, and it can attain the parametric $n^{-1/2}$ rate of convergence to the true ACE even when using data-adaptive estimators of the propensity score and outcome regression that may have convergence rates slower than $n^{-1/2}$. General semiparametric estimation strategies with similar robustness properties have been derived in settings where the causal model is represented as a causal graph with latent confounders [Fulcher et al., 2020, Jung et al., 2021, Bhattacharya et al., 2022]. However, valid causal interpretation of these semiparametric estimators relies on correct specification of the causal model. Furthermore, causal models typically include assumptions that are untestable using the observed data, and which can only be justified using scientific arguments—classic examples are the conditional ignorability assumption in the backdoor model and the exclusion restrictions in the instrumental variable (IV) [Balke and Pearl, 1993, Angrist et al., 1996] and front-door models [Pearl, 1995a].

In some cases, there are multiple plausible causal models identifying a causal effect in a single observed dataset. For example, the data may contain a set of covariates for which conditional ignorability is plausible, and also contain a plausible IV. Evidence factors is an approach to combining hypothesis tests of a common causal null hypothesis under two or more candidate causal models [Rosenbaum, 2010, 2011, Karmakar et al., 2019]. Under certain conditions, evidence factors methodology yields a test that is valid if at least one of the underlying causal models is correct, without knowing which of the models is correct. This is a form of *causal robustness* because the test is robust to misspecifica-

tion of some of the causal models as long as one is correctly specified. This approach allows the analyst to make weaker causal assumptions at the expense of stronger statistical assumptions, since a well-behaved statistical test must be constructed using *each* posited causal model.

In this paper, we propose methods for combining semiparametric theory with evidence factors to produce tests that exhibit both statistical and causal robustness. Our proposed approach is built upon the evidence factors design, where multiple analyses are used to test a common causal null hypothesis using a single dataset. We propose tests based on joint asymptotic normality of multiple asymptotically linear semiparametric estimators, where each estimator is based on a distinct identifying functional derived from a (possibly incorrect) causal model. We show our tests have asymptotically valid type I error rate if at least one of the causal models is correct.

**Advantages of our method:** Our tests have several advantages over existing evidence factors methods, including relaxing some of the conditions required by standard evidence factors designs [Rosenbaum, 2010, 2011, 2021].

(i) Since our tests are based on semiparametric estimators, they possess the types of statistical robustness discussed above.

(ii) We remove the need to demonstrate that the joint distribution of the p-values from multiple tests stochastically dominates the uniform distribution under the null, which is commonly used to demonstrate that the combined p-value from an evidence factors analysis has valid size under the null. Asymptotic validity of our test is guaranteed by joint convergence in distribution of the estimators, which is a consequence of asymptotic linearity of semiparametric estimators.

(iii) Finally, our method does not require that the candidate causal models have non-overlapping sources of bias. In other words, our test is valid even if the assumptions of two or more of the candidate causal models are invalidated by the same source of bias; e.g., the same unmeasured confounder.

The weaker conditions of our proposed approach allow us to readily apply our method to complex settings. We illustrate this with two examples that have not been studied before to the best of our knowledge. In the first example, we consider three candidate causal models: backdoor, front-door, and IV. In the second example, we consider three candidate backdoor models with different adjustment sets. We evaluate the effectiveness of our proposed test using simulations. We then demonstrate our method with two real-world applications. First, we study the effect of smoking on blood glucose levels using data from the Framingham Heart Study [Kannel and Gordon, 1968] by combining analyses from a backdoor, front-door and IV model. Finally, we compare our

methods with evidence factors analysis using the Wisconsin Longitudinal study [Karmakar et al., 2021].

**Other related work:** In addition to the evidence factors work cited earlier, we note that Sun et al. [2021] proposed a multiply robust method for estimating causal effects in a Mendelian randomization setting. Their work is specific to a setting where the candidate models are all IV models. An advantage of our work is that it can be applied in settings where the candidate models are qualitatively distinct. We also note that there is prior research on specification testing for causal models—e.g., Entner et al. [2013] and Shah et al. [2022] proposed tests for conditional ignorability models, Bhattacharya and Nabi [2022] proposed tests for front-door models, and Pearl [1995a] and Wang et al. [2017] proposed tests for IV models. In contrast, we do not aim to test the specification of causal models. Instead, our goal is to test a causal null hypothesis provided that assumptions of at least one of the underlying causal models hold, without knowing which set of assumptions holds.

## 2 MOTIVATING EXAMPLE

We first describe an empirical example to motivate our general theory and methods. We present the results of data analysis for this example in Section 5. We are interested in testing the causal null hypothesis that there is no average causal effect (ACE) of smoking on glucose levels because high glucose levels are a cause of diabetes. We use data from the Framingham Heart Study [Kannel and Gordon, 1968] to test this null hypothesis. The data are observational, and consist of $n = 3477$ fully observed realizations of the data structure $O = \{C, Z, A, M, Y\}$, and we will assume these data are independent and identically distributed from a distribution $P$. In this data, $C$ denotes a set of baseline covariates containing age, sex, BMI, past history of heart disease, and past glucose level; $A$ is binary current smoking status, which is our treatment of interest; $Y$ is glucose level, which is our continuous outcome of interest; $M$ is hypertension, which is our candidate mediator; and $Z$ is past hypertension, which is our candidate IV. We define the ACE of smoking on glucose as $\beta = \mathbb{E}[Y(A = 1)] - \mathbb{E}[Y(A = 0)]$, where $Y(A = 1)$ and $Y(A = 0)$ denote potential outcomes under assignment to smoking and no smoking, respectively. Our causal null hypothesis is $H_0 : \beta = 0$.

Identification of the causal parameter $\beta$ using the distribution of the observed data relies on assumptions encoded in a causal model. Here, as is often the case, there are multiple plausible causal models. Figure 1 shows three plausible causal models for this study using causal directed acyclic graphs (DAGs) [Spirtes et al., 2000, Pearl, 2009]. Each causal DAG only includes the subset of variables important for identification. Solid blue edges represent causal relations that are permitted by the model—i.e., do not violate its identifying assumptions if present in the underlying data

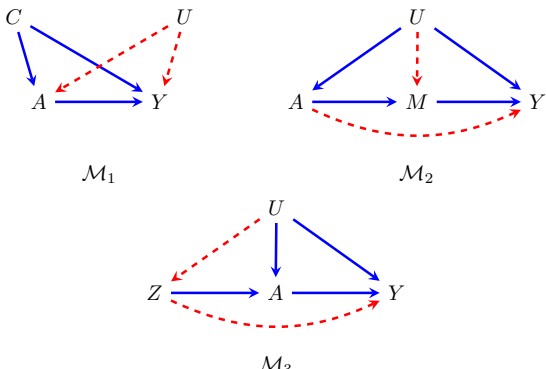

Figure 1: Plausible causal models and violations of their assumptions (shown via red dashed edges). $\mathcal{M}_1$ is a backdoor model, $\mathcal{M}_2$ is a front-door model, and $\mathcal{M}_3$ is an IV model.

generating process—and red dashed edges represent causal relations that are not permitted by the model.

Model $\mathcal{M}_1$ is a model that assumes the smoking-glucose relationship is unconfounded given the observed covariates $C$. Under $\mathcal{M}_1$, the ACE $\beta$ is identified with the observed data parameter $\psi_1(P)$ given by the backdoor formula

$$\psi_{1,P} = \mathbb{E}\left[\mu(1,C) - \mu(0,C)\right], \qquad (1)$$

where $\mu(a,c) := \mathbb{E}(Y \mid A = a, C = c)$ [Robins, 1986, Pearl, 1995a]. For brevity, we will refer to models like $\mathcal{M}_1$ that permit identification via the backdoor formula as backdoor models. Model $\mathcal{M}_2$ is a front-door model that assumes that smoking only impacts glucose through its effect on hypertension, but permits unmeasured common causes of smoking and glucose (but not hypertension). Under $\mathcal{M}_2$, the ACE is identified with the parameter $\psi_{2,P}$ given by

$$\begin{aligned} \psi_{2,P} := \mathbb{E}\{&\mathbb{E}\left[\gamma(M,C) \mid A = 1, C\right] \\ &- \mathbb{E}\left[\gamma(M,C) \mid A = 0, C\right]\}, \end{aligned} \qquad (2)$$

where $\gamma(m,c) := \mathbb{E}[\mu(m,A,c) \mid C = c]$ for $\mu(m,a,c) := \mathbb{E}(Y \mid M = m, A = a, C = c)$ [Pearl, 1995a]. Model $\mathcal{M}_3$ is an IV model that assumes prior hypertension is exogenous and only impacts glucose through its effect on smoking, but permits unmeasured common causes of smoking and glucose (but not previous hypertension). Under $\mathcal{M}_3$, the ACE is identified with $\psi_{3,P}$ given by

$$\psi_{3,P} = \frac{\mathbb{E}(Y \mid Z = 1) - \mathbb{E}(Y \mid Z = 0)}{\mathbb{E}(A \mid Z = 1) - \mathbb{E}(A \mid Z = 0)} \qquad (3)$$

[Balke and Pearl, 1993, Angrist et al., 1996]. We note that each causal model above also includes non-graphical assumptions, such as positivity, for identification to hold. These will be stated in Section 4.

Semiparametric estimators that exhibit robustness to nuisance estimation have been developed for $\psi_{1,P}$, $\psi_{2,P}$, and

$\psi_{3,P}$. For example, the AIPW estimator of $\psi_{1,P}$ [Bang and Robins, 2005] is doubly robust with respect to estimators of the outcome regression and propensity score, the augmented primal IPW estimator of $\psi_{2,P}$ [Fulcher et al., 2020, Bhattacharya et al., 2022] is doubly robust with respect to estimators of the outcome regression and conditional distribution of the mediator, and the empirical plug-in estimator of $\psi_{3,P}$ does not require any nuisance estimators.

If the assumptions in causal model $\mathcal{M}_k$ hold, a semiparametric estimator of the corresponding identified parameter $\psi_{k,P}$ can be used to construct a statistically robust hypothesis test of the causal null hypothesis that $\beta = 0$. For example, if the backdoor model $\mathcal{M}_1$ holds, then a hypothesis test based on the AIPW estimator will have power tending to one as long as either the outcome regression or propensity score estimator is consistent, and will have asymptotically valid type I error rate as long as the outcome regression and propensity score estimators achieve sufficient rates of convergence (which may be slower than $n^{-1/2}$).

If the causal model $\mathcal{M}_k$ fails to hold, a hypothesis test based on $\psi_{k,P}$ may not provide any information about whether $\beta = 0$. Furthermore, it is often the case that some of the assumptions in a causal model do not imply any testable constraints on the observed data distribution. Indeed, in all three models proposed in Figure 1, the absence of the red dashed edges is untestable. The IV assumptions can be falsified via an inequality constraint, but not confirmed [Pearl, 1995b, Wang et al., 2017]. Therefore, the plausibility of causal models typically relies on substantive arguments. In the context of observational studies, such substantive arguments are frequently tenuous. For example, health consciousness is an unmeasured covariate in the Framingham Heart Study that could impact the likelihood of smoking and impact glucose levels through its effect on diet and exercise. If so, the backdoor model $\mathcal{M}_1$ may not hold. It is also possible that smoking impacts glucose through mechanisms other than hypertension, such as reduced likelihood of exercising or reduced appetite, which would invalidate the front-door model $\mathcal{M}_2$. Finally, past history of hypertension may not be exogenous, because diet and exercise may be associated with both hypertension and glucose, which would invalidate the IV model $\mathcal{M}_3$.

To relax the reliance on a single causal model, evidence factors can be used to derive a test of the null hypothesis $H_0 : \beta = 0$ that is valid as long as at least one of $\mathcal{M}_1, \mathcal{M}_2$, or $\mathcal{M}_3$ is true, without knowing which is true. This is a form of *causal robustness*. Evidence factors typically require that the joint distribution of the individual p-values stochastically dominates the uniform distribution under the null. In our approach, asymptotic validity of the test is instead guaranteed by joint convergence in distribution of the estimators, which follows directly from asymptotic linearity of semiparametric estimators. In addition, standard evidence factors analyses require that the source of bias that invalidates one

causal model does not necessarily bias other causal models. For example, the presence of an unmeasured confounder $U$ that also causes past history of hypertension, such as health consciousness, biases $\mathcal{M}_3$ and may bias $\mathcal{M}_1$ as well unless the backdoor paths through $Z$ and $U$ are blocked by $C$. Previous evidence factors literature has used blocking or stratification to preclude such cases [Zhao et al., 2022, Karmakar et al., 2021], which can reduce effective sample size and statistical power. Our proposed approach relaxes this condition and allows one source of bias to potentially invalidate multiple analyses, and adds statistical robustness to the analyses as described above.

# 3 METHOD FOR COMBINING EVIDENCE FACTORS

We propose a new method for combining evidence factors that takes advantage of the asymptotic linearity of influence function-based estimators. We first outline our formal problem setup. Let $\beta$ denote the causal parameter of interest, such as the average causal effect or conditional average causal effect. The causal null hypothesis of interest is $H_0 : \beta = 0$. We assume the observed data consists of $n$ realizations $O_1, \ldots, O_n$ drawn IID from an unknown distribution $P$. We suppose that the analyst is considering $K > 1$ causal models $\mathcal{M}_1, \ldots, \mathcal{M}_K$, and that $\psi_{k,P}$ is an identifying functional for $\beta$ under $\mathcal{M}_k$. That is, if the assumptions of $\mathcal{M}_k$ are true, then $\beta = \psi_{k,P}$, which further implies that $H_0$ holds if and only if $\psi_{k,P} = 0$. Hence, if at least one of the causal models $\mathcal{M}_1, \ldots, \mathcal{M}_K$ is true, then under the causal null hypothesis $H_0$, at least one of $\psi_{1,P}, \ldots, \psi_{K,P}$ is zero. Equivalently, if at least one of the $K$ causal models is true, then $H_0$ implies that $\prod_{k=1}^{K} \psi_{k,P} = 0$. This motivates our approach to testing $H_0$. We note that the reverse implication is not necessarily true; we discuss this further later in this section.

For each $k$, we suppose we can construct an asymptotically linear estimator $\psi_{k,n}$ of $\psi_{k,P}$ with influence function $\phi_{k,P}$ under statistical conditions $\mathcal{C}_k$, meaning $\psi_{k,n} - \psi_{k,P} = \mathbb{P}_n \phi_{k,P} + o_P(n^{-1/2})$, where $\mathbb{P}_n f = \frac{1}{n} \sum_{i=1}^{n} f(O_i)$. Here, $\phi_{k,P}$ may depend on $P$, and is assumed to satisfy $\mathbb{E}(\phi_{k,P}) = 0$ and $\mathbb{E}(\phi_{k,P}^2) < \infty$. The statistical conditions $\mathcal{C}_k$ typically include rates of convergence and complexity constraints for nuisance estimators such as outcome regression or propensity score estimators, as well as constraints on $P$ such as finite moments or semi-parametric or parametric modeling assumptions.

## 3.1 JOINT DISTRIBUTION OF ASYMPTOTICALLY LINEAR ESTIMATORS

Asymptotic linearity implies the marginal convergence result $n^{1/2}(\psi_{k,n} - \psi_{k,P}) \to_d N(0, \sigma_{k,P}^2)$ for $\sigma_{k,P}^2 := \mathbb{E}(\phi_{k,P}^2)$, which can be used to construct asymptotically

valid Wald-style confidence intervals for $\psi_{k,P}$. A natural estimator of the asymptotic variance $\sigma_{k,P}^2$ is given by $\sigma_{k,n}^2 := \mathbb{P}_n \phi_{k,n}^2$, where $\phi_{k,n}$ is an estimator of the influence function $\phi_{k,P}$. This is known as the *influence function-based variance estimator* [van der Vaart, 2000]. However, asymptotic linearity is stronger than marginal convergence. In particular, by the multivariate central limit theorem, asymptotic linearity of any finite collection of estimators implies *joint* convergence in distribution of the estimators. Denoting $\boldsymbol{\psi}_P := (\psi_{1,P}, \ldots, \psi_{K,P})'$ and $\boldsymbol{\psi}_n := (\psi_{1,n}, \ldots, \psi_{K,n})'$ as vectors of the true and estimated parameters, respectively, and $\boldsymbol{\phi}_P := (\phi_{1,P}, \ldots, \phi_{K,P})'$ as the vector of influence functions, if all the statistical conditions $\mathcal{C}_1, \ldots, \mathcal{C}_K$ hold, then

$$\boldsymbol{\psi}_n - \boldsymbol{\psi}_P = \mathbb{P}_n \boldsymbol{\phi}_P + o_P(n^{-1/2}).$$

This implies $n^{1/2}(\boldsymbol{\psi}_n - \boldsymbol{\psi}_P) \to_d N_K(\mathbf{0}, \boldsymbol{\Sigma}_P)$, where $\boldsymbol{\Sigma}_P$ is defined as

$$\begin{pmatrix} \mathbb{E}(\phi_{1,P}^2) & \mathbb{E}(\phi_{1,P}\phi_{2,P}) & \cdots & \mathbb{E}(\phi_{1,P}\phi_{K,P}) \\ \mathbb{E}(\phi_{2,P}\phi_{1,P}) & \mathbb{E}(\phi_{2,P}^2) & \cdots & \mathbb{E}(\phi_{2,P}\phi_{K,P}) \\ \vdots & \vdots & \ddots & \vdots \\ \mathbb{E}(\phi_{K,P}\phi_{1,P}) & \mathbb{E}(\phi_{K,P}\phi_{2,P}) & \cdots & \mathbb{E}(\phi_{K,P}^2) \end{pmatrix}.$$

We can estimate $\boldsymbol{\Sigma}_P$ using the influence function-based covariance estimator $\boldsymbol{\Sigma}_n$ by estimating $\mathbb{E}(\phi_{j,P}\phi_{k,P})$ with $\mathbb{P}_n \phi_{j,n}\phi_{k,n}$. If $\boldsymbol{\Sigma}_n \to_P \boldsymbol{\Sigma}_P$ and $\boldsymbol{\Sigma}_P$ is invertible, it follows that $n^{1/2}\boldsymbol{\Sigma}_n^{-1/2}(\boldsymbol{\psi}_n - \boldsymbol{\psi}_P) \to_d N_K(\mathbf{0}, \boldsymbol{I}_K)$, where $I_K$ is the $K \times K$ identity matrix and $\boldsymbol{\Sigma}_n^{-1/2}$ is the inverse of the matrix square root of $\boldsymbol{\Sigma}_n$.

## 3.2 TESTS OF THE IMPLIED NULL BASED ON JOINT ASYMPTOTIC NORMALITY

We propose using the joint convergence implied by asymptotic linearity to derive tests of the null hypothesis that $\psi_{k,P} = 0$ for at least one $k$. By the delta method we have

$$n^{1/2}\left(\prod_{k=1}^{K}\psi_{k,n} - \prod_{k=1}^{K}\psi_{k,P}\right) \to_d N\left(0, \boldsymbol{\gamma}_P' \boldsymbol{\Sigma}_P \boldsymbol{\gamma}_P\right),$$

where $\boldsymbol{\gamma}_P := (\gamma_{1,P}, \ldots, \gamma_{K,P})'$ for $\gamma_{k,P} := \prod_{j \neq k} \psi_{j,P}$. We recall that if at least one of the causal models is correctly specified, then the causal null hypothesis $H_0$ implies that $\prod_{k=1}^{K}\psi_{k,P} = 0$, which then implies that

$$T_n := n^{1/2}\left(\boldsymbol{\gamma}_n' \boldsymbol{\Sigma}_n \boldsymbol{\gamma}_n\right)^{-1/2}\prod_{k=1}^{K}\psi_{k,n} \to_d N(0,1).$$

Therefore, a two-sided test of $H_0$ with asymptotically valid type I error rate is given by rejecting at level $\alpha$ if $|T_n| > q_{1-\alpha/2}$, where $q_p$ denotes the $p$th quantile of a standard normal distribution. The following result formally demonstrates that this proposed test has asymptotically valid type I error rate.

**Theorem 1.** *Suppose that for each $k \in \{1, \ldots, K\}$, $\psi_{k,n}$ is an asymptotically linear estimator of $\psi_{k,P}$ with influence function $\phi_{k,P}$, $\prod_{k=1}^{K}\psi_{k,P} = 0$, and $\boldsymbol{\Sigma}_n \to_P \boldsymbol{\Sigma}_P$, where $\boldsymbol{\gamma}_P' \boldsymbol{\Sigma}_P \boldsymbol{\gamma}_P > 0$. Then $P\left(|T_n| > q_{1-\alpha/2}\right) \longrightarrow \alpha$.*

A proof of Theorem 1 is given in the Appendix. Theorem 1 is stated in terms of the statistical properties of the test, and we now elaborate on how this relates to our goal of developing tests with causal model and statistical robustness. Theorem 1 implies that if at least one of the causal models $\mathcal{M}_1, \ldots, \mathcal{M}_K$ is true and *all* of the statistical conditions $\mathcal{C}_1, \ldots, \mathcal{C}_K$ implying asymptotic linearity of the estimators $\psi_{1,n}, \ldots, \psi_{K,n}$ are true, then the test that rejects the causal null hypothesis $H_0 : \beta = 0$ when $|T_n| > q_{1-\alpha/2}$ has asymptotic size $\alpha$. Hence, increasing $K$ relaxes the causal conditions at the expense of stronger statistical conditions. By using semiparametric estimators rather than estimators based on parametric models, we increase the statistical robustness in conditions $\mathcal{C}_1, \ldots, \mathcal{C}_K$.

We now briefly comment on some conditions under which we may not get precise type I error control, and justify why these situations may not be considered problematic in practice. First, if more than one $\psi_{k,P}$ equals zero, then $\gamma_P = 0$, which implies that $\gamma_P' \Sigma_P \gamma_P = 0$. Hence, our method only yields precise type I error control when exactly one of $\psi_{1,P}, \ldots, \psi_{K,P}$ equals zero. If two or more equal zero, then the rate of convergence of $\prod_{k=1}^K \psi_{k,n}$ is faster than $n^{-1/2}$, and so our test will be asymptotically conservative. This will be illustrated in simulations in Section 4 discussed further in Section 6. Briefly, this might occur when the null hypothesis is true and the analyst has successfully specified two or more causal models correctly. In practice, however, we expect a scenario in which the analyst is able to specify more than one model correctly to be exceptionally rare—often our concern is if even a single model has been correctly specified. Readers interested in learning more about developing tests in such scenarios may also refer to Miles and Chambaz [2021] for a test developed in a separate context that has better power in the special case of $K = 2$ and diagonal $\Sigma_P$. To our knowledge, no such test yet exists in the general case.

Second, we note that $\gamma_P' \Sigma_P \gamma_P$ may equal 0 if $\mathbb{E}(\phi_{k,P}^2) = 0$. Hence, precise type I error control using our method also relies on the variances $\mathbb{E}(\phi_{k,P}^2)$ being positive when $\psi_{k,P} = 0$. In some cases, $\psi_{k,P} = 0$ implies that $\mathbb{E}(\phi_{k,P}^2) = 0$. If this happens, our test may again be asymptotically conservative. For example, suppose the null hypothesis $H_0$ is the strong causal null hypothesis that there is no causal effect of a binary treatment $A$ on an outcome $Y$ for any unit in the population. Under the backdoor model, $H_0$ implies that $\psi := \mathbb{E}\{[\mu(1, C) - \mu(0, C)]^2\} = 0$, where $\mu(1, c) - \mu(0, c)$ is the conditional average treatment effect. When $\psi = 0$, its efficient influence function is 0 [Levy et al., 2021]. However, since the strong null hypothesis implies the weak null hypothesis that the ACE equals zero, the problem can be avoided in this case by testing the weak null instead.

Finally, $\gamma_P' \Sigma_P \gamma_P$ may equal 0 if two or more of the influence functions are linearly dependent under the null hypothesis. Fortunately, this can be checked by the researcher prior

to using the method.

The next result provides conditions under which the power of the test goes to one under fixed alternatives.

**Theorem 2.** *Suppose that for each $k \in \{1, \ldots, K\}$, $\psi_{k,n} \to_P \psi_{k,P}$, where $\prod_{k=1}^K \psi_{k,P} \neq 0$, and $\Sigma_n = \mathrm{O}_P(1)$. Then $P\left(|T_n| > q_{1-\alpha/2}\right) \longrightarrow 1$.*

A proof of Theorem 2 is provided in the Appendix. The conditions of Theorem 2 are substantially weaker than those of Theorem 1. In particular, Theorem 2 only requires consistency of the estimators, which for doubly robust estimators can hold as long as at least one nuisance estimator is consistent.

We note that $\prod_{k=1}^K \psi_{k,P} \neq 0$ requires that each $\psi_{k,P} \neq 0$. If $\mathcal{M}_k$ is a correct causal model, then $\psi_{k,P} \neq 0$ if and only if $\beta \neq 0$. However, if $\mathcal{M}_k$ is invalid, then $\psi_{k,P}$ does not necessarily have any correspondence with $\beta$, and hence $\psi_{k,P}$ may equal 0 even if $\beta \neq 0$. Hence, the power of the proposed test may not converge to one under certain alternatives even if at least one of $\mathcal{M}_1, \ldots, \mathcal{M}_K$ is true and all of the statistical conditions $\mathcal{C}_1, \ldots, \mathcal{C}_K$ are true. This phenomenon will be illustrated in numerical studies in Section 4. It appears that developing a consistent test in situations where $\beta \neq 0$ but some $\psi_{k,P} = 0$ would require being able to determine which models are invalid, which as discussed above is typically not possible. However, in some cases, even when $\mathcal{M}_k$ is invalid, $\psi_{k,P} = 0$ is an "unlikely" event when $\beta \neq 0$ in the sense that it requires exact cancellations of certain causal effects. This is related to the *faithfulness* assumption in DAGs [Spirtes et al., 2000], which states that (conditional) independence between variables under $P$ can always be attributed to the structure of the causal graph. In causal graphical selection, $P$ is often assumed to be faithful with respect to a causal graph with the justification that unfaithful distributions are rare [Spirtes et al., 2000]. If 1) the distribution $P$ is faithful and 2) $\psi_{k,P} = 0$ if and only if $Y \perp\!\!\!\perp A \mid R$, where $R$ denotes other observed variables appearing in $\psi_{k,P}$, then $\beta \neq 0$ implies that $\psi_{k,P} \neq 0$. Condition 2) holds in, for example, some linear Gaussian models. An example of a causal model violating faithfulness is shown in the Appendix.

# 4 PRACTICAL APPLICATIONS OF THE GENERAL METHOD

As noted in Section 3, our method can be applied to any set of causal models as long as we can construct asymptotically linear estimators of each $\psi_{k,P}$. Recent developments in semiparametric theory allow us to do this for any identified query of the ACE given a causal graph with unmeasured confounders [Bhattacharya et al., 2022, Jung et al., 2021].

We highlight two important examples here: (i) three qualitatively distinct causal models—backdoor, front-door, and IV,

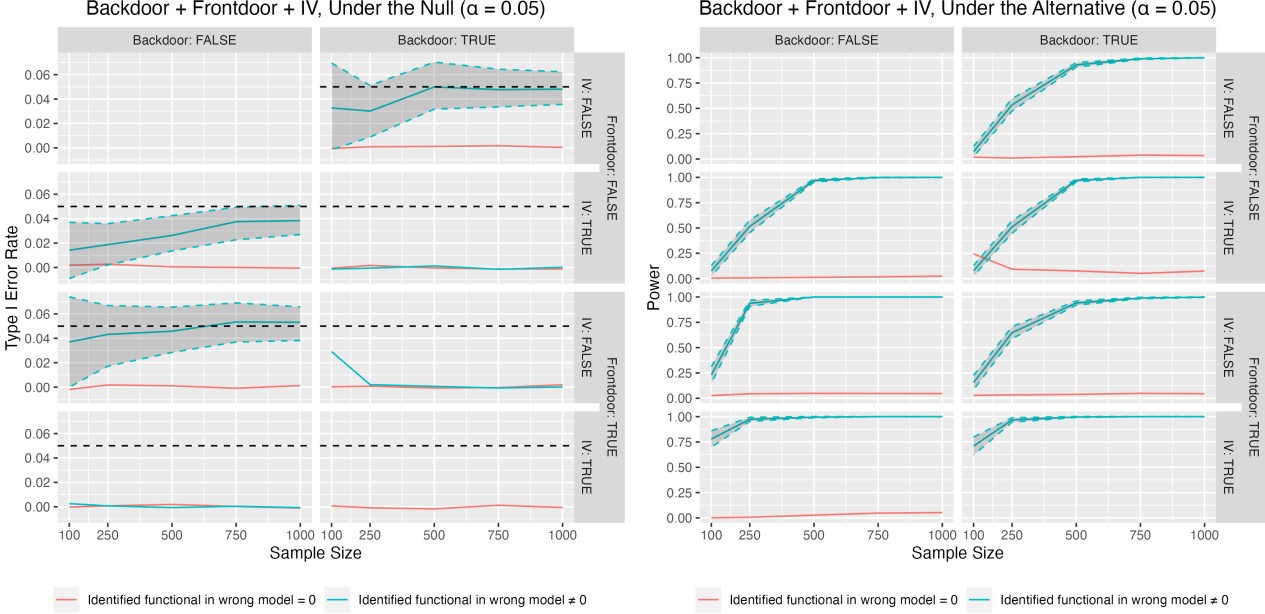

Figure 2: Size (left) and power (right) of the test as a function of sample size when at least one of backdoor, front-door, or IV are true. Panel labels indicate which model(s) are correct (TRUE) and incorrect (FALSE).

and (ii) multiple plausible backdoor models. We assess the performance of our proposed test using numerical studies in both examples. In Section 5 we demonstrate an application of (i) to the Framingham Heart Study, as highlighted in our motivating example. We also compare our method with prior evidence factors work using the Wisconsin Longitudinal Study using two distinct IV models and a backdoor model.

### 4.1 BACKDOOR, FRONT-DOOR, AND IV MODELS

We return to the three candidate causal models introduced in Section 2 and displayed in Figure 1: the backdoor, front-door, and IV models. Before describing the numerical study, we provide additional details about the causal models and estimators. We are interested in testing the weak causal null hypothesis $H_0 : \beta = \mathbb{E}[Y(A = 1) - Y(A = 0)] = 0$. Causal model $\mathcal{M}_1$ is the backdoor model. In addition to SUTVA and consistency, the assumptions of $\mathcal{M}_1$ are: (i) $Y(A = a) \perp\!\!\!\perp A \mid C$ for $a \in \{0, 1\}$ (conditional ignorability), and (ii) $0 < \pi(C) < 1$ almost surely for $a \in \{0, 1\}$ for $\pi(c) := P(A = 1 \mid C = c)$ (positivity). Under these conditions, $\beta = \psi_{1,P}$ defined in (1). The nonparametric efficient influence function of $\psi_{1,P}$ is $\phi_{1,P} = \phi_{1,P}^\circ - \psi_{1,P}$, where $\phi_{1,P}^\circ(y, a, c)$ is given by

$$\{y - \mu(a, c)\} \left\{ \frac{a - \pi(c)}{\pi(c)[1 - \pi(c)]} \right\} + \{\mu(1, c) - \mu(0, c)\}.$$

The AIPW estimator [Bang and Robins, 2005] is an asymptotically linear estimator of $\psi_{1,P}$ with influence function

$\phi_{1,P}$ under doubly robust conditions on estimators $\mu_n$ and $\pi_n$ of $\mu$ and $\pi$, respectively.

Causal model $\mathcal{M}_2$ is the front-door model. The key assumptions of $\mathcal{M}_2$ are: (i) $Y(A = a, M = m) = Y(M = m)$ for $a, m \in \{0, 1\}$ (no direct effect of treatment on the outcome); (ii) $Y(M = m) \perp\!\!\!\perp M(A = a) \mid C$ for $a, m \in \{0, 1\}$ (conditional ignorability of the mediator-outcome relationship); (iii) $M(A = a) \perp\!\!\!\perp A \mid C$ for $a \in \{0, 1\}$ (conditional ignorability of the treatment-mediator relationship); (iv) $Y(M = m) \perp\!\!\!\perp M \mid A, C$; and (v) $0 < P(A = a, M = m \mid C) < 1$ almost surely for each $a, m \in \{0, 1\}$ (positivity). Unobserved confounding between $A$ and $Y$ is permitted. Under these conditions, $\beta = \psi_{2,P}$ defined in (2). The nonparametric efficient influence function $\phi_{2,P}(y, m, a, c)$ of $\psi_{2,P}$ is

$$\frac{\alpha(m \mid 1, c) - \alpha(m \mid 0, c)}{\alpha(m \mid a, c)} \{y - \mu(m, a, c)\}$$
$$+ \left\{ \frac{a - \pi(c)}{\pi(c)[1 - \pi(c)]} \right\} \{\gamma(m, c) - \tau(a, c)\}$$
$$+ \{\eta(1, a, c) - \eta(0, a, c)\} - \psi_{2,P},$$

where $\alpha(m \mid a, c) := P(M = m \mid A = a, C = c)$, $\eta(a_0, a, c) := \mathbb{E}[\mu(M, a, c) \mid A = a_0, C = c]$, and $\tau(a, c) := \mathbb{E}[\eta(a, A, c) \mid C = c]$. The augmented primal IPW estimator of $\psi_{2,P}$ [Fulcher et al., 2020, Bhattacharya et al., 2022] is asymptotically linear with influence function $\phi_{2,P}$ under double robust conditions on estimators of the sets $\{\pi, \mu\}$ and $\{\alpha\}$.

Finally, causal model $\mathcal{M}_3$ is an IV model. The key assumptions of $\mathcal{M}_3$ are: (i) $Y(Z = z) \perp\!\!\!\perp Z$, for $z \in \{0, 1\}$ (ran-

domized instrument); (ii) $Y(Z = z, A = a) = Y(A = a)$ for each $a, z \in \{0, 1\}$ (no direct effect of the instrument on the outcome); (iii) $P(A(Z = 0) = 1, A(Z = 1) = 0) = 0$ (monotonicity); (iv) $\mathbb{E}[A(Z = 1) - A(Z = 0)] \neq 0$ (non-null effect of the instrument on treatment); (v) $\text{Var}\{Y(A = 1) - Y(A = 0)\} = 0$ (homogeneity); and (vi) $0 < P(Z = 1) < 0$ (positivity). Unobserved confounding of the treatment-outcome relationship is again permitted. Under these conditions, $\beta = \psi_{3,P}$ defined in (3). We note that without the homogeneity assumption, $\psi_{3,P}$ is identified with the ACE among compliers, so we use it here to identify our actual target $\beta$. The nonparametric efficient influence function $\phi_{3,P}(y, a, z)$ of $\psi_{3,P}$ is

$$[\{y - \mu(z)\}\{\pi(1) - \pi(0)\} - \{a - \pi(z)\}\{\mu(1) - \mu(0)\}]$$
$$\times \frac{z/\zeta - (1-z)/(1-\zeta)}{\{\pi(1) - \pi_0(0)\}^2},$$

where $\mu(z) := \mathbb{E}(Y \mid Z = z)$, $\pi(z) := P(A = 1 \mid Z = z)$, and $\zeta := P(Z = 1)$. Since $A$ and $Z$ are binary, an asymptotically linear estimator of $\psi_{3,P}$ with influence function $\phi_{3,P}$ can be constructed by replacing the conditional expectations in the definition of $\psi_{3,P}$ given in (3) with empirical conditional expectations.

We note that it is possible that $\mathcal{M}_1$, $\mathcal{M}_2$, and $\mathcal{M}_3$ are invalidated by a common source of bias. For example, if $Z$ has a direct effect on $Y$, this invalidates both the IV model $\mathcal{M}_3$ and the backdoor model $\mathcal{M}_1$ (if $Z$ is not in the adjustment set $C$). Unlike previous evidence factors analyses [Karmakar et al., 2021, Zhao et al., 2022], we do not alter the adjustment sets nor impose any restrictions on the order of analyses to prevent the source of bias of one model from invalidating others.

In the first numerical study, we consider testing the causal null hypothesis $H_0 : \beta = 0$ against the two-sided alternative using our proposed test with $K = 3$ using the three causal models $\mathcal{M}_1$, $\mathcal{M}_2$, and $\mathcal{M}_3$ defined above. We consider settings where the assumptions of all causal models hold, where the assumptions of two of the models hold, and where the assumptions of just one model holds. For each setting, we consider data-generating distributions where the identified functional in the incorrect models is 0 or is different from 0 because we expect this to impact the rejection rate of the test, as discussed in Section 3.2. To violate the assumptions of $\mathcal{M}_1$, we either include unmeasured confounders or adjust for colliders. To violate the assumptions of $\mathcal{M}_2$, we either include an effect of $A$ on $Y$ not mediated through $M$ or include unmeasured confounding between $A$ and $M$ or between $M$ and $Y$. To violate the assumptions of $\mathcal{M}_3$, we include a direct effect of $Z$ on $Y$, include unmeasured confounding between $Z$, $A$, and $Y$, or violate monotonicity. To simultaneously violate the assumptions of $\mathcal{M}_1$ and $\mathcal{M}_2$, we use a common source of bias: a direct effect of $Z$ on $Y$. The full details of the data-generating processes for each setting are in the Appendix.

For each data-generating distribution, we simulate data under the null and alternative hypotheses for sample sizes $n \in \{100, 250, 500, 750, 1000\}$. For each simulated dataset, we use our proposed test with the estimators and influence functions described above. We estimate outcome regression and propensity score functions using generalized additive models. For each setting and sample size, we conduct 1000 simulations and record the fraction of the time that our test rejected the null hypothesis at level $\alpha = 0.05$.

Figure 2 displays the size and power of the test as a function of sample size under the different settings. The results are consistent with our expectations based on the theory of Section 3.2. Under the null (left panel of Figure 2) the size of the test converges to $\alpha = 0.05$ when two of the causal models are wrong and both identified functionals in the wrong models are not zero. The size is close to zero when more than one of the causal models are correct or when the identified functional in the wrong model is zero. This is because, as discussed in Section 3.2, our test is conservative when more than one $\psi_{k,P}$ equals zero. Under the alternative (right panel of Figure 2), the power of the test converges rapidly to 1 in all cases when the identified functionals in the wrong model are not zero. The test has low power when identified functional in the wrong model equals zero as discussed in Section 3.2.

We also consider our proposed test with $K = 2$ using all three pairs of models: $\mathcal{M}_1$ and $\mathcal{M}_2$, $\mathcal{M}_1$ and $\mathcal{M}_3$, and $\mathcal{M}_2$ and $\mathcal{M}_3$. The simulation results for these settings can be found in the Appendix, and again align with our theoretical expectations.

### 4.2 MULTIPLE BACKDOOR MODELS

In the second example, we consider $K = 3$ backdoor models with different adjustment sets. Figure 4 displays the true causal graph. The adjustment set of the first backdoor model is $\{C_1, C_2, C_3, C_4\}$. This model is correct because this set satisfies the backdoor criterion with respect to $A$ and $Y$. The second adjustment set is $\{C_1, C_3\}$, and the third adjustment set is $\{C_1, C_4\}$, so both of these adjustment sets are invalid because they omit the confounder $C_2$. As long as the common source of bias shared by multiple analyses does not affect all candidate models, then our approach can still be valid, which is again one of the stated advantages of our method over standard evidence factors designs.

We use our proposed test with three AIPW estimators with the three different adjustment sets. We use generalized additive models to estimate the outcome regression and propensity score. Figure 3 displays the results of the second numerical study. The results are consistent with our expectations. Under the null, the size of the test converges to $\alpha = 0.05$, but is slightly anti-conservative for $n = 250$. Under the alternative, the power of the test is close to 1 for all $n$ be-

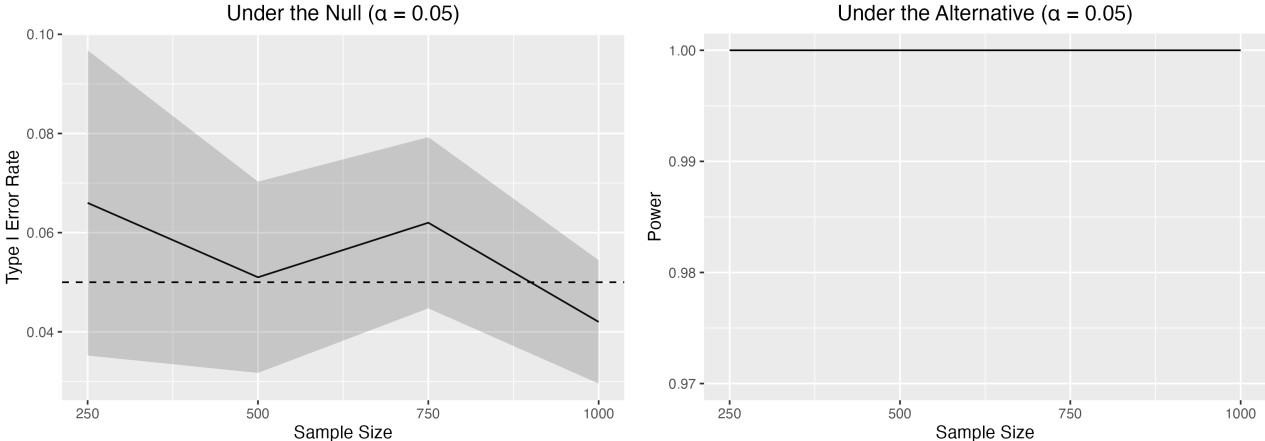

Figure 3: Size (left) and power (right) of the test as a function of sample size for the second numerical study.

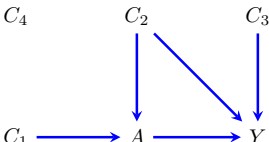

Figure 4: True causal DAG for the backdoor models.

cause in this case, the identified functionals in the backdoor models with invalid adjustment sets are not zero.

# 5 REAL DATA APPLICATIONS

In this section we evaluate our methods using two real-world studies. The first is the Framingham Heart Study as introduced in our example in Section 2. The second is the Wisconsin Longitudinal Study that has been analyzed using classical evidence factors methods by Karmakar et al. [2021] and thus allows us to compare our methods with prior work.

## 5.1 FRAMINGHAM HEART STUDY

We first use our methods to test the effect of smoking on glucose levels using the Framingham Heart Study [Kannel and Gordon, 1968]. We use the backdoor, front-door, and IV models defined in Sections 2 and 4 as our candidate models. Our treatment $A$ is a binary indicator of current smoking status and our outcome $Y$ is a continuous measure of blood glucose level. We adjust for baseline covariates $C$ containing age, sex, BMI, past history of heart disease, and past glucose level in the backdoor model. We propose hypertension as a candidate mediator $M$ for the front-door model, and past history of hypertension as a candidate instrumental variable $Z$ for the IV model. We estimate the ACE in each candidate model using the methods described in Section 4.

Table 1 displays the estimates, 95% confidence intervals,

and p-values from the tests of the null hypothesis of zero ACE using each causal model individually. The tests based on the backdoor and front-door models fail to reject the null hypothesis that smoking status has no effect on glucose levels. The test based on the IV model rejects the null hypothesis at significance level $0.05$ and produces an estimated ACE less than zero, suggesting that smoking reduces glucose levels. However, these results all rely on validity of the single causal model on which they are based. The joint test proposed here is valid if any of the three causal models is valid and returns a p-value of $0.68$. Hence, we do not find evidence of a statistically significant causal effect of smoking on glucose levels.

| Method | $\widehat{\text{ACE}}$ (95% CI) | p-value |
|:---:|:---:|:---:|
| Backdoor | 0.32 (-1.2, 1.8) | 0.67 |
| Front-door | -0.038 (-0.090, 0.014) | 0.15 |
| IV | -47.7 (-62.8, -32.6) | $6.5 \times 10^{-10}$ |

Table 1: Results from the analysis of the effect of smoking on glucose from the Framingham heart study.

## 5.2 WISCONSIN LONGITUDINAL STUDY

We next evaluate our method with the Wisconsin Longitudinal Study (WLS) dataset from the `R` package `blockingChallenge` [Karmakar, 2018]. We compare our methods and results to evidence factors analysis for this data [Karmakar et al., 2021].

The WLS data contains a sample of 4450 male students who completed high school in Wisconsin in 1957. The binary exposure of interest is whether the student attended a Catholic high school, and the outcome is income in 1974. Karmakar et al. [2021] considered three causal models: (1) an IV model using whether the student's family resided

in an urban or rural area during high school as an instrument; (2) an IV model using whether the student's family was Catholic as an instrument and urban/rural residence as a covariate; and (3) a backdoor model adjusting for both urban/rural residence and Catholic religion as covariates. Each model also included IQ score prior to high school, father's and mother's education, parents' income, father's occupation score, and occupational prestige score as covariates. Letting $\beta$ be the ACE of attending a Catholic school on income, we use the methods of Karmakar et al. [2021] to test the null hypothesis that $\beta = 0$ versus the alternative that $\beta \neq 0$ in these three models, and combine these three causal models using evidence factors methodology. We assume that at least one model is correct, so we combine the individual p-values from the evidence factors analysis by taking the maximum of the three.

We apply our proposed test with the three causal models described above with slight modifications using the methods described in Section 4. For the two IV models, we do not adjust for any covariates, and for the backdoor model, we adjust for all covariates excluding the two candidate IVs.

| Urban IV | Catholic IV | Backdoor | Combined |
|---|---|---|---|
| **Evidence Factors Analysis** | | | |
| $< 0.0001$ | 0.0084 | 0.0098 | 0.0098 |
| **Asymptotic Joint Test** | | | |
| $3.3 \times 10^{-14}$ | 0.0094 | 0.0004 | 0.0950 |

Table 2: Results comparing of our method to evidence factors analysis in analyzing the effect of Catholic schooling on wages from the Wisconsin longitudinal study.

Table 2 displays the p-values from the three individual tests and the combined test using the evidence factors methodology and our methodology. While all individual p-values are statistically significant at the $0.01$ level, our combined p-value is not. This is because the three individual p-values using our proposed models are positively correlated, while the p-values using the evidence factors methods are nearly independent of each other under the null by carefully constructing each evidence factor analysis. In particular, the estimated correlation between the AIPW estimator and the IV estimators from the Catholic religion and urban/rural IV models are 0.42 and 0.22, respectively. Therefore, whereas the combined p-value from the evidence factors analysis simply takes the maximum among the three p-values, our method takes into account the correlations among the three tests. Our method produces a valid test even if the individual p-values are positively correlated and does not require particular causal models to make p-values from each analysis nearly independent under the null.

# 6  CONCLUSION

Many of the assumptions of causal models in the context of observational data are strong and empirically untestable. It is desirable to use methods that are as robust as possible in such settings in order to relax the strength of the assumptions. In this paper, we proposed a method of testing a causal null hypothesis in the presence of several candidate causal models that provides both statistical and causal robustness. Our test is valid if at least one of the proposed causal models is correct, without knowing which one is correct. Furthermore, our test is based on semiparametric estimators, which possess desirable statistical robustness properties. Our methods also relax standard evidence factors conditions in two ways: we remove the requirement that non-overlapping biases invalidate the causal models, and we do not need to show the distribution of the p-values from each factor dominates the uniform distribution under the null. This has allowed us to apply our method to new settings for which evidence factors have not yet been developed. We expect there are applications of our work to additional new settings, as well as extensions to causal sensitivity analysis.

The relaxation of the second condition comes at the cost of statistical power when more than one causal model is correct. Some evidence factor analyses allow researchers to assume $J \geq 2$ of the $K$ causal models are correct, without knowing which $J$ causal models are correct [Rosenbaum, 2010, 2011]. The resulting combined test is more powerful as $J$ increases, at the expense of stronger conditions and less robustness to invalid causal models. In particular, if the practitioner assumes that $J > 1$ models are correct, when in truth fewer than $J$ are correct, then the resulting test has invalid type I error rate. Here, we only considered the situation where $J = 1$, and if the number of true causal models exceeds one, our test is valid but tends to be conservative. Extending our approach to settings where $J \geq 2$ models are correct is an important area of future research.

Our theory covers the case where the number of causal models $K$ is fixed, and we were primarily focused on the situation where $K$ is relatively small. Another interesting area of future research is to quantify the trade-offs in robustness and power as a function of $K$ and the dependence between the estimators in each model. We expect that increasing $K$ typically comes with a reduction in power. However, we also believe that qualitatively distinct causal models, such as the backdoor, front-door, and IV models considered here, leads to less power reduction than qualitatively similar models, such as multiple backdoor models, because the power of the combined test is lower when the individual p-values are positively correlated.

Finally, we focused here on testing causal null hypotheses because testing is the main focus of the evidence factors literature, and is an important aspect of causal inference across various disciplines such as epidemiology [Swanson

et al., 2018], political science [Eggers et al., 2023], and economics Angrist and Kuersteiner [2011]. However, as with evidence factors, we expect that our tests can be inverted to construct robust confidence sets. This too is an important topic of future research.

## Acknowledgements

The authors gratefully acknowledge support from NSF grant 2113171 (TW) and the helpful feedback of four anonymous reviewers.

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

# Statistical and Causal Robustness for Causal Null Hypothesis Tests (Supplementary Material)

**Junhui Yang**[1,*]      **Rohit Bhattacharya**[2,*]      **Youjin Lee**[3]      **Ted Westling**[1]

[1]Department of Mathematics and Statistics, University of Massachusetts Amherst, Amherst, Massachusetts, USA
[2]Department of Computer Science, Williams College, Williamstown, Massachusetts, USA
[3]Department of Biostatistics, Brown University, Providence, Rhode Island, USA
[*]Equal contributors

## A  PROOF OF THEOREMS

***Proof of Theorem 1.*** Asymptotic linearity of $\boldsymbol{\psi}_n$ implies that $n^{1/2}(\boldsymbol{\psi}_n - \boldsymbol{\psi}_P) \to_d N(\mathbf{0}, \boldsymbol{\Sigma}_P)$, where $\boldsymbol{\Sigma}_P := \mathbb{E}[\boldsymbol{\phi}_P \boldsymbol{\phi}_P']$ is the asymptotic covariance matrix. Let $h : \mathbb{R}^K \to \mathbb{R}$ be defined pointwise as $h(x_1, x_2, \ldots, x_K) := \prod_{k=1}^K x_k$. Then $h$ is a continuously differentiable function with $\frac{\partial h}{\partial x_k}(x_1, x_2, \ldots, x_K) = \prod_{j \neq k} x_j$ for each $k$. Denoting the gradient mapping of $h$ by $\nabla h$, by the delta method,

$$n^{1/2} \left[ h(\boldsymbol{\psi}_n) - h(\boldsymbol{\psi}_P) \right] = n^{1/2} \left( \prod_{k=1}^K \psi_{k,n} - \prod_{k=1}^K \psi_{k,P} \right) \to_d N(0, \sigma_P^2)$$

for

$$\sigma_P^2 := \nabla h(\boldsymbol{\psi}_P)' \boldsymbol{\Sigma}_P \nabla h(\boldsymbol{\psi}_P) = \boldsymbol{\gamma}_P' \boldsymbol{\Sigma}_P \boldsymbol{\gamma}_P.$$

Since $\boldsymbol{\Sigma}_n \to_P \boldsymbol{\Sigma}_P$ by assumption, by the continuous mapping theorem [Mann and Wald, 1943], $(\boldsymbol{\gamma}_n' \boldsymbol{\Sigma}_n \boldsymbol{\gamma}_n)^{1/2} \to_P (\boldsymbol{\gamma}_P' \boldsymbol{\Sigma}_P \boldsymbol{\gamma}_P)^{1/2} = \sigma_P$, which is positive by assumption. Therefore, since $\prod_{k=1}^K \psi_{k,P} = 0$,

$$n^{1/2} (\boldsymbol{\gamma}_n' \boldsymbol{\Sigma}_n \boldsymbol{\gamma}_n)^{-1/2} \prod_{k=1}^K \psi_{k,n} \to_d N(0, 1).$$

The result follows.        $\square$

***Proof of Theorem 2.*** By the continuous mapping theorem [Mann and Wald, 1943], $\prod_{k=1}^K \psi_{k,n} \to_P \prod_{k=1}^K \psi_{k,P} \neq 0$, and $\boldsymbol{\gamma}_n \to_P \boldsymbol{\gamma}_P$. Since $\boldsymbol{\Sigma}_n = O_P(1)$, $(\boldsymbol{\gamma}_n' \boldsymbol{\Sigma}_n \boldsymbol{\gamma}_n)^{1/2} = O_P(1)$ as well. Therefore, $|T_n| \to_P +\infty$, which yields the result.    $\square$

## B  EXAMPLE OF A CAUSAL MODEL VIOLATING FAITHFULNESS

Figure 5 shows an example of a causal model that violates faithfulness due to exact cancellation and where $\beta \neq 0$ but $\psi_{k,P} = 0$ when applying the backdoor formula with observed covariates. In this example, each variable is equal to a linear function of its direct causes and an independent noise term; e.g., $Y = 2A - 2U + 4C + \epsilon_Y$. Here, the causal null $H_0$ that $A$ has no causal effect on $Y$ is false – the causal effect of $A$ on $Y$ is the coefficient 2. This distribution violates faithfulness because $A$ and $Y$ are not d-separated given $C$ [Pearl, 2009], but nevertheless it turns out that $Y \perp\!\!\!\perp A \mid C$. To see this, we use Wright's rules of path analysis (assuming all variables are standardized) [Wright, 1921] to find that $\mathrm{Cor}(A, Y \mid C) = -2 \times 1 + 2 = 0$. Since $Y$ is given by a linear combination of its causes, this implies $Y \perp\!\!\!\perp A \mid C$. Since the conditional independence $Y \perp\!\!\!\perp A \mid C$ does not correspond to a property of the graph, it violates faithfulness. Furthermore, while the backdoor model with conditioning set $C$ is false due to the unblocked backdoor path through $U$, the observed data parameter identified by the backdoor model is given by $\psi_{1,P} = \mathrm{Cor}(A, Y \mid C) = 0$ as above. Hence, $\psi_{1,P} = 0$ even though $\beta \neq 0$, which is due to the violation of faithfulness.

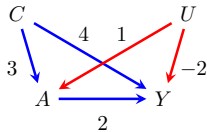

Figure 5: A causal model violating faithfulness.

## C  ADDITIONAL DETAILS FOR SIMULATIONS STUDIES

Here we provide details for the data-generating processes for the simulation studies presented in Section 4. The coefficient $\beta$ was set to 0 under the null and set to 10 under alternatives. We define $\text{expit}(x) := 1/[1 + \exp(-x)]$ for $x \in \mathbb{R}$. Throughout, "$\text{Bern}(p)$" is shorthand for the Bernoulli distribution with probability $p$, "$\text{Unif}(a, b)$" is shorthand for the continuous uniform distribution on the interval $[a, b]$, and $N(\mu, \sigma^2)$ is shorthand for the normal distribution with mean $\mu$ and variance $\sigma^2$.

### C.1  BACKDOOR, FRONT-DOOR, AND IV MODELS

(a)

(b)

(c)

(d)

(e)

(f)

(g)

(h)

(i)

(j)

(k)

Figure 6: Causal DAGs for the data-generating distribution for the simulation with backdoor, front-door, and IV models. Violation of assumptions is shown via solid black edges.

We begin with data-generating processes for the simulation study combining the backdoor, front-door, and IV models, the results of which are shown in Figure 2 and discussed in Section 4. Figure 6 shows the causal DAGs for this simulation. Figure 6(a) shows the causal DAG in the setting where all three models are valid, which was used to generate the lines in

the bottom right panels under the null and alternative of Figure 2. The precise data-generating process for this setting is as follows. We first generate

$$U \sim \text{Unif}(-2, 2)$$
$$C_i \sim \text{Unif}(-2, 2), \text{ for } i = 1, 2, 3, 4$$
$$Z \sim \text{Bern}(0.5).$$

We also define

$$\pi(c_1, c_2, c_3) = \text{expit} \left\{ c_1 + \text{expit}(c_2) + \sin(c_3) \right\}.$$

We then simulate $\bar{A}(1) \sim \text{Bern}(\pi(C_1, C_2, C_3))$ and $\bar{A}(0) \sim \text{Bern}(1 - \pi(C_1, C_2, C_3))$. To make the monotonicity assumption hold for the IV model, we then convert all defiers to compliers by setting $A(1) = 1$ and $A(0) = 0$ if $\bar{A}(1) = 0$ and $\bar{A}(0) = 1$, and setting $A(1) = \bar{A}(1)$ and $A(0) = \bar{A}(0)$ otherwise. The observed treatment A is then defined as $A = A(Z)$. Finally, we set

$$M \sim \text{Bern} \left( \text{expit}\{5A - 1 + C_2\} \right)$$
$$Y \sim N \left( \beta M + 3U + 2\sqrt{|C_1|} + \sin(C_4), 1 \right).$$

Figure 6(b) shows the causal DAG in the setting where the front-door and IV models are valid, but the backdoor model is invalid due to an unblocked path from $A$ to $Y$ through $U$. This DAG was used to generate the both lines in the bottom left panels under the null and alternative of Figure 2. The data-generating process for this setting when the identified backdoor functional is not zero under the null and alternative is the same as that described for (a) above, but we change $\pi$ to

$$\pi(c_1, c_2, c_3, u) = \text{expit} \left\{ c_1 + \text{expit}(c_2) + \sin(c_3) + u \right\}.$$

The data-generating process for the setting when the identified backdoor functional equals zero under the null is the same as that described for (a) above, but we change the equations for $\pi$ and $Y$ to

$$\pi(c_1, c_2, c_3, u) = \text{expit} \left\{ c_1 + \text{expit}(c_2) + \sin(c_3) - u \right\}$$
$$Y \sim N \left( \beta M + U + 2\sqrt{|C_1|} + \sin(C_4), 1 \right).$$

The data-generating process for the setting when the identified backdoor functional equals zero under the alternative is the same as that described for (a) above, but we change the equations for $\pi$, $M$, and $Y$ to

$$\pi(c_1, c_2, z, u) = \text{expit} \left\{ -0.5 + 5z + c_1 + \text{expit}(c_2) - 0.97u \right\}$$
$$M \sim \text{Bern} \left( \text{expit}\{2A - 1 + C_2\} \right)$$
$$Y \sim N \left( \beta M + 5U - 2\sqrt{|C_1|} + \sin(C_4), 1 \right).$$

Since $U$ has an effect on both $A$ and $Y$ but is not in the adjustment set for the backdoor model, the backdoor model is invalid.

Figure 6(c) shows the causal DAG in the setting where the backdoor and front-door models are valid, but the IV model is invalid due to a direct effect of $Z$ on $Y$. This DAG was used to generate the line corresponding to "Identified functional in wrong model $\neq 0$" in the second-from-bottom right panel under the null of Figure 2. The data-generating process for this setting is the same as that described for (a) above, but we change the equation for $Y$ to

$$Y \sim N \left( \beta M + U + 2\sqrt{|C_1|} + \sin(C_4) + 2Z, 1 \right).$$

Since $Z$ now has a direct effect on $Y$, the IV model is invalid.

To simulate data where the front-door and backdoor models are valid, but the IV model is invalid (second-from-bottom right panels of Figure 2) under the null when the identified functional in the IV model equals 0 and under the alternative, we make the IV model invalid by violating the monotonicity assumption. This violation does not have a graphical visualization, so it is not displayed in Figure 6. The equations for $U$, $C$, $Z$, and $\pi$ are as described for setting (a) above. We then simulate

$A(1) \sim \text{Bern}(\pi(C_1, C_2, C_3))$ and $A(0) \sim \text{Bern}(1 - \pi(C_1, C_2, C_3))$, and we set $A = A(Z)$. Finally, we change the equations for $M$ and $Y$ to

$$M \sim \text{Bern}\left(\text{expit}\{\alpha_1 A + \alpha_2 I\{A(0) < A(1)\}A - 1 + C_2\}\right)$$
$$Y \sim N\left(\beta M + U + 2\sqrt{|C_1|} + \sin(C_4), 1\right).$$

Here, we set $\alpha_1 = 5$ and $\alpha_2 = -3$ under the null, we set $\alpha_1 = 5$ and $\alpha_2 = -2.838$ under the alternative if the identified IV functional equals zero, and we set $\alpha_1 = 2$ and $\alpha_2 = 3$ under the alternative if the identified IV functional is not zero. Since there are "defiers" for whom $A(0) = 1$ but $A(1) = 0$, the IV model is invalid.

Figure 6(d) shows the causal DAG in the setting where the front-door model is valid, but the backdoor model is invalid due to an unblocked path from $A$ to $Y$ through $V$ and the IV model is invalid due to an unblocked path from $Z$ to $Y$ through $U$. This DAG was used to generate the line corresponding to "Identified functional in wrong model $\neq 0$" in the second-from-bottom left panel under the null of Figure 2. The data-generating process for this setting is the same as that described for (a) above, but we add $V \sim \text{Unif}(-2, 2)$ and change the equations for $Z$, $\pi$, $M$ and $Y$ to

$$Z \sim \text{Bern}\left(\text{expit}\{2 + 2U\}\right)$$
$$\pi(c_1, c_2, c_3, v) = \text{expit}\left\{c_1 + \text{expit}(c_2) + \sin(C_3) + v\right\}$$
$$M \sim \text{Bern}\left(\text{expit}\{2A - 1 + C_2\}\right)$$
$$Y \sim N\left(\beta M + 2U + V + 2\sqrt{|C_1|} + \sin(C_4), 1\right).$$

Since $V$ has an effect on both $A$ and $Y$, but is not in the adjustment set for the backdoor model, the backdoor model is invalid. Since $U$ has an effect on both $Z$ and $Y$, but is not in the adjustment set for the IV model, the IV model is invalid.

To simulate data where the front-door model is valid but the backdoor and IV models are invalid (second-from-bottom left panels of Figure 2) under the null when "Identified functional in wrong model = 0" and under the alternative when "Identified functional in wrong model = 0", we make the IV model invalid by violating the monotonicity assumption. This violation does not have a graphical visualisation, so it is not displayed in Figure 6. The backdoor model is invalid due to an unblocked path from A to Y through U. The equations for $U$, $C$, $Z$, and $Y$ are as described for setting (a) above. We change the equation for $\pi$ to

$$\pi(c_1, c_2, c_3, u) = \text{expit}(c_1 + \text{expit}(c_2) + \sin(c_3) + u).$$

We then simulate $A(1) \sim \text{Bern}(\pi(C_1, C_2, C_3, U))$ and $A(0) \sim \text{Bern}(1 - \pi(C_1, C_2, C_3, U))$, and we set $A = A(Z)$. Finally, we change the equations for $M$ and $Y$ to

$$M \sim \text{Bern}\left(\text{expit}\{\alpha_1 A + \alpha_2 I\{A(0) < A(1)\}A - 1 + C_2\}\right)$$
$$Y \sim N\left(\beta M + U + 2\sqrt{|C_1|} + \sin(C_4), 1\right).$$

Here, we set $\alpha_1 = 5$ and $\alpha_2 = -3$ under the null, and we set $\alpha_1 = 5$ and $\alpha_2 = -2.63$ under the alternative. Since $U$ has an effect on both $A$ and $Y$, but is not in the adjustment set for the backdoor model, the backdoor model is invalid. Since there are "defiers" for whom $A(0) = 1$ but $A(1) = 0$, the IV model is invalid.

Figure 6(e) shows the causal DAG in the setting where the front-door model is valid, but the backdoor model is invalid due to an unblocked path from $A$ to $Y$ through $Z$ and the IV model is invalid due to a direct effect of $Z$ on $Y$. This DAG was used to generate the line corresponding to "Identified functional in wrong model $\neq 0$" in the second-from-bottom left panel under the alternative of Figure 2. The data-generating process for this setting is the same as that described for (a) above, but we change the equations for $\pi$, $M$, and $Y$ to

$$\pi(c_1, c_2, c_3, u) = \text{expit}\left\{c_1 + \text{expit}(c_2) + \sin(c_3)\right\}$$
$$M \sim \text{Bern}\left(\text{expit}\{2A - 1 + C_2\}\right)$$
$$Y \sim N\left(\beta M + 3U + 2\sqrt{|C_1|} + \sin(C_4) + 2Z, 1\right).$$

Since $Z$ has an effect on both $A$ and $Y$, but is not in the adjustment set for the backdoor model, the backdoor model is invalid. Since $Z$ has a direct effect on $Y$, the IV model is invalid.

Figure 6(f) shows the causal DAG in the setting where the backdoor and IV models are valid, but the front-door model is invalid due to a direct effect of $A$ on $Y$. This DAG was used to generate the line corresponding to "Identified functional in

wrong model $= 0$" in the third-from-bottom right panels under the null and alternative of Figure 2. The data-generating process for this setting is the same as that described for (a) above, but we change the equation for $Y$ to

$$Y \sim N\left(\beta A + 3U + 2\sqrt{|C_1|} + \sin(C_4), 1\right).$$

Since $A$ now has a direct effect on $Y$, the front-door model is invalid.

Figure 6(g) shows the causal DAG in the setting where the backdoor and IV models are valid, but the front-door model is invalid due to an unblocked path from $M$ to $Y$ through $U$. This DAG was used to generate the line corresponding to "Identified functional in wrong model $\neq 0$" in the third-from-bottom right panels under the null and alternative of Figure 2. The data-generating process for this setting is the same as that described for (a) above, but we change the equation for $M$ to

$$M \sim \text{Bern}\left(\text{expit}\{3A - 1 + C_2 + U\}\right).$$

Since $U$ now has an effect on both $M$ and $Y$ but is not in the adjustment set for the front-door model, the front-door model is invalid.

Figure 6(h) shows the causal DAG in the setting where the IV model is valid, but the backdoor model is invalid due to controlling for the collider $C_5$ and the front-door model is invalid due to an unblocked path from $M$ to $Y$ through $U$ and because $M$ does not fully mediate the effect of $A$ on $Y$. This DAG was used to generate the line corresponding to "Identified functional in wrong model $\neq 0$" in the third-from-bottom left panel under the null of Figure 2. The data-generating process for this setting is the same as that described for (a) above, but we change the equations for $\pi$, $M$, and $Y$ to

$$\pi(c_1, c_2, c_3) = \text{expit}\left\{c_4 + \text{expit}(c_2) + \sin(c_3)\right\}$$
$$M \sim \text{Bern}(\text{expit}\{5A - 1 + C_2 + 2U\})$$
$$Y \sim N\left(\beta M + U + \sin(C_4), 1\right)$$

and we simulate $C_5 \sim N(3A - Y, 1)$. Since $C_5$ is a $A$-$Y$ collider and it is adjusted for in the backdoor model, the backdoor model is invalid. Since $U$ has an effect on both $M$ and $Y$ but is not in the adjustment set for the front-door model, the front-door model is invalid.

Figure 6(i) shows the causal DAG in the setting where the IV model is valid, but the backdoor model is invalid due to an unblocked path from $A$ to $Y$ through $U$ and the front-door model is invalid due to an unblocked path from $M$ to $Y$ through $U$. This DAG was used to generate the line corresponding to "Identified functional in wrong model $= 0$" in the third-from-bottom left panel under the null and the line corresponding to "Identified functional in wrong model $\neq 0$" in the third-from-bottom left panel under the alternative of Figure 2. The data-generating process for this setting is the same as that described for (a) above, but we change the equations for $\pi$ and $M$ to

$$\pi(c_1, c_2, c_3, u) = \text{expit}\left\{c_1 + \text{expit}(c_2) + \sin(c_3) + u\right\}$$
$$M \sim \text{Bern}\left(\text{expit}\{3A - 1 + C_2 + U\}\right).$$

Since $U$ has an effect on both $A$ and $Y$, but is not in the adjustment set for the backdoor model, the backdoor model is invalid. Since $U$ has an effect on both $M$ and $Y$, but is not included in the adjustment set for the font-door model, the front-door model is invalid.

Figure 6(j) shows the causal DAG in the setting where the IV model is valid, but the backdoor model is invalid due to controlling for the collider $C_5$ and the front-door model is invalid due to an unblocked path from $A$ to $M$ through $U$ and because $M$ does not fully mediate the effect of $A$ on $Y$. This DAG was used to generate the line corresponding to "Identified functional in wrong model $= 0$" in the third-from-bottom left panel under the alternative of Figure 2. The data-generating process for this setting is the same as that described for (a) above, but we change the equations for $\pi$, $M$, and $Y$ to

$$\pi(c_1, c_2, c_3, u) = \text{expit}\left\{c_4 + \sin(c_3) - u\right\}$$
$$M \sim \text{Bern}\left\{\text{expit}(5A - 1 + C_2 - 2U)\right\}$$
$$Y \sim N\left(\beta M - 5\sin(C_4), 1\right)$$

and we simulate $C_5 \sim N(-2A - 5Y, 1)$. Since $C_5$ is a $A$-$Y$ collider and it is adjusted for in the backdoor model, the backdoor model is invalid. Since $U$ has an effect on both $A$ and $M$ but is not in the adjustment set for the front-door model, the front-door model is invalid.

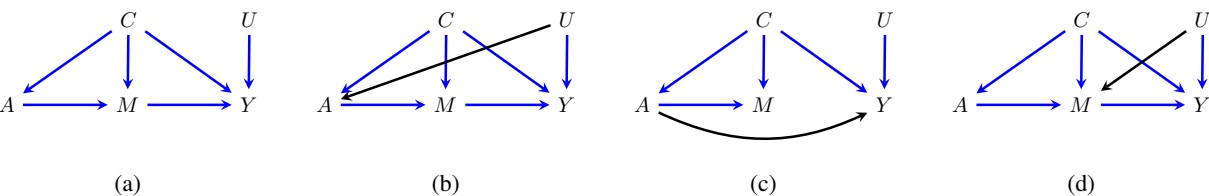

(a)       (b)       (c)       (d)

Figure 7: Causal DAGs for the data-generating distribution for the simulation with backdoor and front-door models. Violation of assumptions is shown via solid black edges.

Figure 6(k) shows the causal DAG in the setting where the backdoor model is valid, but the front-door model is invalid due to an unblocked path from $M$ to $Y$ through $U$ and the IV model is invalid due to a direct effect of $Z$ on $Y$. This DAG was used to generate the line corresponding to "Identified functional in wrong model $\neq 0$" in the top right panel under the null of Figure 2. The data-generating process for this setting is the same as that described for (a) above, but we change the equations for $M$ and $Y$ to

$$M \sim \mathrm{Bern}\left(\mathrm{expit}\{2A - 1 + C_2 + U\}\right)$$
$$Y \sim N\left(\beta M - 3U + 2\sqrt{|C_1|} + \sin(C_4) + 2Z, 1\right).$$

Since $U$ has an effect on both $M$ and $Y$ but is not in the adjustment set for the front-door model, the front-door model is invalid. Since $Z$ has a direct effect on $Y$, the IV model is invalid. We note that $Z$ is included in the adjustment set for the backdoor model, since otherwise there would be an unblocked path from $A$ to $Y$ through $Z$.

To simulate data where the backdoor model is valid but the front-door and IV models are invalid (top right panels of Figure 2) under the null when "Identified functional in wrong model = 0" and under the alternative when "Identified functional in wrong model $\neq 0$", we make the IV model invalid by violating the monotonicity assumption. As above, this violation does not have a graphical visualisation, so it is not displayed in Figure 6. The front-door model is invalid due to an unblocked path from M to Y through U. The equations for $U$, $C$, $Z$, and $\pi$ are as described for setting (a) above. We then simulate $A(1) \sim \mathrm{Bern}(\pi(C_1, C_2, C_3))$ and $A(0) \sim \mathrm{Bern}(1 - \pi(C_1, C_2, C_3))$, and we set $A = A(Z)$. We also change the equations for $M$ and $Y$ to

$$M \sim I\{A(0) < A(1)\}\mathrm{Bern}\left(\mathrm{expit}\{5A - 1 + C_2 + U\}\right) + I\{A(0) \geq A(1)\}\mathrm{Bern}\left(\mathrm{expit}\{2A - 1 + C_2 + U\}\right)$$
$$Y \sim N\left(\beta M - 3U + 2\sqrt{|C_1|} + \sin(C_4), 1\right).$$

Since $U$ has an effect on both $M$ and $Y$, but is not in the adjustment set for the front-door model, the front-door model is invalid. Since there are "defiers" for whom $A(0) = 1$ but $A(1) = 0$, the IV model is invalid.

To simulate data where the backdoor model is valid but the front-door and IV models are invalid in the top right panel under the alternative of Figure 2 when "Identified functional in wrong model = 0", we make the IV model invalid by violating the monotonicity assumption. As above, this violation does not have a graphical visualisation, so it is not displayed in Figure 6. The front-door model is invalid due to a direct effect of A on Y. The equations for $U$, $C$, $Z$, and $\pi$ are as described for setting (a) above. We then simulate $A(1) \sim \mathrm{Bern}(\pi(C_1, C_2, C_3))$ and $A(0) \sim \mathrm{Bern}(1 - \pi(C_1, C_2, C_3))$, and we set $A = A(Z)$. We also change the equations for $M$ and $Y$ to

$$M \sim I\{A(0) < A(1)\}\mathrm{Bern}\left(\mathrm{expit}\{2A - 1 + C_2\}\right) + I\{A(0) \geq A(1)\}\mathrm{Bern}\left(\mathrm{expit}\{5A - 1 + C_2\}\right)$$
$$Y \sim N\left(\beta A + 3U + 2\sqrt{|C_1|} + \sin(C_4), 1\right).$$

## C.2   BACKDOOR AND FRONT-DOOR MODELS

We next present the data-generating processes for the simulation study combining the backdoor and front-door models, the results of which are shown in Figure 10. Figure 7 shows the causal DAGs for this simulation. Figure 7(a) shows the causal DAG in the setting where both models are valid, which was used to generate the lines in the bottom right panels under the

null and alternative of Figure 10. The precise data-generating process for this setting is as follows. We generate

$$U \sim \text{Unif}(-2, 2)$$
$$C_i \sim \text{Unif}(-2, 2), \text{ for } i = 1, 2, 3, 4$$
$$A \sim \text{Bern}\left(\text{expit}\left\{C_1 + \text{expit}(C_2) + \sin(C_3)\right\}\right)$$
$$M \sim \text{Bern}\left(\text{expit}\{2A - 1 + C_2\}\right)$$
$$Y \sim N\left(\beta M + 2U + 2\sqrt{|C_1|} + \sin(C_4), 1\right).$$

Figure 7(b) shows the causal DAG in the setting where the front-door model is valid, but the backdoor model is invalid due to an unblocked path from $A$ to $Y$ through $U$. This DAG was used to generate both lines in the bottom left panels under the null and alternative of Figure 10. The data-generating process for this setting when "Identified functional in wrong model $\neq$ 0" under the null and under the alternative is the same as that described for (a) above, but we change the formula for $A$ to

$$A \sim \text{Bern}\left(\text{expit}\left\{C_1 + \text{expit}(C_2) + \sin(C_3) + U\right\}\right).$$

The data-generating process for this setting when "Identified functional in wrong model = 0" under the null is the same as that described for (a) above, but we change the equations for $A$, $M$, and $Y$ to

$$A \sim \text{Bern}\left(\text{expit}\left\{C_1 + \text{expit}(C_2) + \sin(C_3) - 0.05U\right\}\right)$$
$$M \sim \text{Bern}\left(\text{expit}\{5A - 1 + C_2\}\right)$$
$$Y \sim N\left(\beta M + 0.05U + 2\sqrt{|C_1|} + \sin(C_4), 1\right).$$

The data-generating process for this setting when "Identified functional in wrong model = 0" under the alternative is the same as that described for (a) above, but we change the equations for $A$, $M$, and $Y$ to

$$A \sim \text{Bern}\left(\text{expit}\left\{C_1 - \text{expit}(C_2) - \sin(C_3) + 0.6U\right\}\right)$$
$$M \sim \text{Bern}\left(\text{expit}\{0.37A - 1 + C_2\}\right)$$
$$Y \sim N\left(\beta M - 0.9U + 2\sqrt{|C_1|} + \sin(C_4), 1\right).$$

Since $U$ has an effect on both $A$ and $Y$ but is not in the adjustment set for the backdoor model, the backdoor model is invalid.

Figure 7(c) shows the causal DAG in the setting where the backdoor model is valid, but the front-door model is invalid due to a direct effect of $A$ on $Y$. This DAG was used to generate the line corresponding to "Identified functional in wrong model = 0" in the top right panels under the null and alternative of Figure 10. The data-generating process for this setting is the same as that described for (a) above, but we change the formula for $Y$ to

$$Y \sim N\left(\beta A + 2U + 2\sqrt{|C_1|} + \sin(C_4), 1\right).$$

Since $A$ has a direct effect on $Y$, the front-door model is invalid.

Figure 7(d) shows the causal DAG in the setting where the backdoor model is valid, but the front-door model is invalid due to an unblocked path from $M$ to $Y$ through $U$. This DAG was used to generate the line corresponding to "Identified functional in wrong model $\neq$ 0" in the top right panels under the null and alternative of Figure 10. The data-generating process for this setting is the same as that described for (a) above, but we change the formula for $M$ to

$$M \sim \text{Bern}\left(\text{expit}\{2A - 1 + C_2 + U\}\right).$$

Since $U$ has an effect on both $M$ and $Y$, but is not included in the adjustment set for the front-door model, the front-door model is invalid.

## C.3 BACKDOOR AND IV MODELS

We next present the data-generating processes for the simulation study combining the backdoor and IV models, the results of which are shown in Figure 11. Figure 8 shows the causal DAGs for this simulation. Figure 8(a) shows the causal DAG in

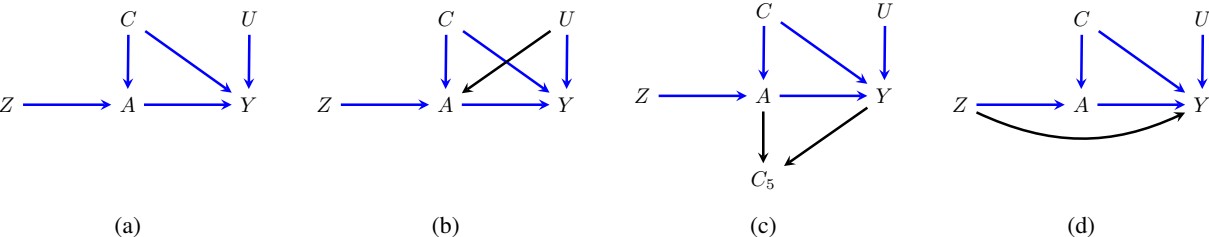

Figure 8: $K = 2$ with backdoor and IV models. Violation of assumptions is shown via solid black edges.

the setting where both models are valid, which was used to generate the lines in the bottom right panels under the null and alternative of Figure 11. The precise data-generating process for this setting is as follows. We first generate

$$U \sim \text{Unif}(-2, 2)$$
$$C_i \sim \text{Unif}(-2, 2), \text{ for } i = 1, 2, 3, 4$$
$$Z \sim \text{Bern}(0.5).$$

We also define

$$\pi(c_1, c_2, c_3) = \text{expit}\left\{c_1 + \text{expit}(c_2) + \sin(c_3)\right\}.$$

We then simulate $\bar{A}(1) \sim \text{Bern}(\pi(C_1, C_2, C_3))$ and $\bar{A}(0) \sim \text{Bern}(1 - \pi(C_1, C_2, C_3))$. As above, to make the monotonicity assumption hold for the IV model, we then convert all defiers to compliers by setting $A(1) = 1$ and $A(0) = 0$ if $\bar{A}(1) = 0$ and $\bar{A}(0) = 1$, and setting $A(1) = \bar{A}(1)$ and $A(0) = \bar{A}(0)$ otherwise. The observed treatment A is then defined as $A = A(Z)$. Finally, we set

$$Y \sim N\left(\beta A + 2U + 2\sqrt{|C_1|} + \sin(C_4), 1\right).$$

Figure 8(b) shows the causal DAG in the setting where the IV model is valid, but the backdoor model is invalid due to an unblocked path from $A$ to $Y$ through $U$. This DAG was used to generate the line corresponding to "Identified functional in wrong model = 0" in the bottom left panel under the null of Figure 11. The data-generating process for this setting is the same as that described for (a) above, but we change the formula for $\pi$ to

$$\pi(c_1, c_2, c_3, u) = \text{expit}\left\{c_1 + \text{expit}(c_2) + \sin(c_3) + u\right\}.$$

Since $U$ has an effect on both $A$ and $Y$ but is not in the adjustment set for the backdoor model, the backdoor model is invalid.

Figure 8(c) shows the causal DAG in the setting where the IV model is valid, but the backdoor model is invalid due to controlling for the collider $C_5$. This DAG was used to generate the lines corresponding to "Identified functional in wrong model $\neq 0$" in the bottom left panels under the null and alternative of Figure 11 as well as the line corresponding to "Identified functional in wrong model = 0" in the bottom left panel under the alternative of Figure 11. The data-generating process for this setting when "Identified functional in wrong model $\neq 0$" under the null is the same as that described for (a) above, but we change the equations for $\pi$ and $Y$ to

$$\pi(c_1, c_2, c_3) = \text{expit}\left\{c_4 + \text{expit}(c_2) + \sin(c_3)\right\}$$
$$Y \sim N\left(\beta A + 2U + \sin(C_4), 1\right).$$

We then simulate $C_5$ as

$$C_5 \sim N\left(2A + Y, 1\right).$$

The data-generating process for this setting when "Identified functional in wrong model $\neq 0$" under the alternative is the same as that described for (a) above, but we change the equations for $\pi$ and $Y$ to

$$\pi(c_1, c_2, c_3) = \text{expit}\left\{c_4 + \text{expit}(c_2) + \sin(c_3)\right\}$$
$$Y \sim N\left(\beta A + 2U + \sin(C_4), 1\right).$$

We then simulate $C_5$ as

$$C_5 \sim N\left(A + Y, 1\right).$$

The data-generating process for this setting when "Identified functional in wrong model = 0" under the alternative is the same as that described for (a) above, but we change the equations for $\pi$ and $Y$ to

$$\pi(c_1, c_2, c_3) = \text{expit}\left\{c_4 + \text{expit}(c_2) + \sin(c_3)\right\}$$
$$Y \sim N\left(\beta A - 3U - \sin(C_4), 1\right).$$

We then simulate $C_5$ as

$$C_5 \sim N\left(0.6A + 2Y, 1\right).$$

Since $C_5$ is a collider and is included in the adjustment set for the backdoor model, the backdoor model is invalid.

Figure 8(d) shows the causal DAG in the setting where the backdoor model is valid, but the IV model is invalid due to a direct effect of $A$ on $Y$. This DAG was used to generate the line corresponding to "Identified functional in wrong model $\neq$ 0" in the top left panel under the null of Figure 11. The data-generating process for this setting is the same as that described for (a) above, but we change the equation for $Y$ to

$$Y \sim N\left(\beta A + 2U + 2\sqrt{|C_1|} + \sin(C_4) + 2Z, 1\right).$$

Since $Z$ has a direct effect on $Y$, the IV model is invalid.

To simulate data where the backdoor model is valid but the IV model is invalid (top right panels of Figure 11) under the null when the identified functional in the IV model equals 0 and under both cases for the alternative, we make the IV model invalid by violating the monotonicity assumption. As above, this violation does not have a graphical visualisation, so it is not displayed in Figure 8. The equations for $U$, $C$, $Z$, and $\pi$ are as described for setting (a) above. We then simulate $A(1) \sim \text{Bern}(\pi(C_1, C_2, C_3))$ and $A(0) \sim \text{Bern}(1 - \pi(C_1, C_2, C_3))$, and we set $A = A(Z)$. We also change the equation for $Y$ to

$$Y \sim N\left(\beta_1 A + \beta_2 I\{A(0) > A(1)\}A + 2U + 2\sqrt{|C_1|} + \sin(C_4), 1\right)$$

Here, we set $\beta_1 = 0$ and $\beta_2 = 0$ under the null, we set $\beta_1 = 5.75$ and $\beta_2 = 4.25$ under the alternative if the identified IV functional equals zero, and we set $\beta_1 = 10$ and $\beta_2 = -8$ under the alternative if the identified IV functional is not zero. Since there are "defiers" for whom $A(0) = 1$ but $A(1) = 0$, the IV model is invalid.

## C.4   FRONT-DOOR AND IV MODELS

We next present the data-generating processes for the simulation study combining the front-door and IV models, the results of which are shown in Figure 12. Figure 9 shows the causal DAGs for this simulation. Figure 9(a) shows the causal DAG in the setting where both models are valid, which was used to generate the lines in the bottom right panels under the null and alternative of Figure 12. The precise data-generating process for this setting is as follows. We first generate

$$U \sim \text{Unif}(-2, 2)$$
$$C_i \sim \text{Unif}(-2, 2), \text{ for } i = 1, 2, 3, 4$$
$$Z \sim \text{Bern}(0.5).$$

We also define

$$\pi(c_1, c_2, c_3, u) = \text{expit}\left\{c_1 + \text{expit}(c_2) + \sin(c_3) + u\right\}.$$

We then simulate $\bar{A}(1) \sim (\pi(C_1, C_2, C_3, U))$ and $\bar{A}(0) \sim (1 - \pi(C_1, C_2, C_3, U))$. To make the monotonicity assumption hold for the IV model, we then convert all defiers to compliers by setting $A(1) = 1$ and $A(0) = 0$ if $\bar{A}(1) = 0$ and $\bar{A}(0) = 1$, and setting $A(1) = \bar{A}(1)$ and $A(0) = \bar{A}(0)$ otherwise. The observed treatment A is then defined as $A = A(Z)$. Finally, we set

$$M \sim (\text{expit}\{5A - 1 + C_2\})$$
$$Y \sim N\left(\beta M + 3U + 2\sqrt{|C_1|} + \sin(C_4), 1\right).$$

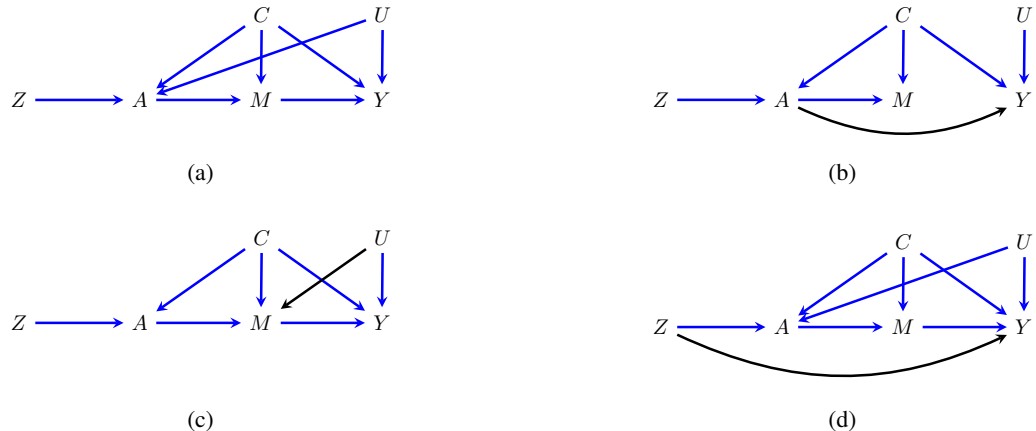

Figure 9: Causal DAGs for the data-generating distribution for the simulation with front-door and IV models. Violation of assumptions is shown via solid black edges.

Figure 9(b) shows the causal DAG in the setting where the IV model is valid, but the front-door model is invalid due to a direct effect of $A$ on $Y$. This DAG was used to generate the line corresponding to "Identified functional in wrong model $=$ 0" in the bottom left panels under the null and alternative of Figure 12. The data-generating process for this setting is the same as that described for (a) above, but we change the equations for $\pi$ and $Y$ to

$$\pi(c_1, c_2, c_3) = \text{expit} \left\{ c_1 + \text{expit}(c_2) + \sin(c_3) \right\}$$
$$Y \sim N \left( \beta A + 3U + 2\sqrt{|C_1|} + \sin(C_4), 1 \right).$$

Since $A$ has a direct effect on $Y$, the front-door model is invalid.

Figure 9(c) shows the causal DAG in the setting where the IV model is valid, but the front-door model is invalid due to an unblocked path from $M$ to $Y$ through $U$. This DAG was used to generate the line corresponding to "Identified functional in wrong model $\neq$ 0" in the bottom left panels under the null and alternative of Figure 12. The data-generating process for this setting is the same as that described for (a) above, but we change the equations for $\pi$ and $M$ to

$$\pi(c_1, c_2, c_3) = \text{expit} \left\{ c_1 + \text{expit}(c_2) + \sin(c_3) \right\}$$
$$M \sim \text{Bern} \left( \text{expit}\{3A - 1 + C_2 + U\} \right).$$

Since $U$ has an effect on both $M$ and $Y$ but is not in the adjustment set for the front-door model, the front-door model is invalid.

Figure 9(d) shows the causal DAG in the setting where the front-door model is valid, but the IV model is invalid due to direct effect of $Z$ on $Y$. This DAG was used to generate the line corresponding to "Identified functional in wrong model $\neq$ 0" in the top right panels under the null and alternative of Figure 12. The data-generating process for this setting is the same as that described for (a) above, but we change the equation for $Y$ to

$$Y \sim N \left( \beta M + 3U + 2\sqrt{|C_1|} + \sin(C_4) + 2Z, 1 \right).$$

Since $Z$ has a direct effect on $Y$, the IV model is invalid.

To simulate data where the front-door model is valid but the IV model is invalid (top right panels of Figure 12) under the null and alternative when the identified functional in the IV model equals 0, we make the IV model invalid by violating the monotonicity assumption. As above, this violation does not have a graphical visualisation, so it is not displayed in Figure 9. The equations for $U$, $C$, $Z$, and $\pi$ are as described for setting (a) above. We then simulate $A(1) \sim (\pi(C_1, C_2, C_3, U))$ and $A(0) \sim (1 - \pi(C_1, C_2, C_3, U))$, and we set $A = A(Z)$. We change the equations for $M$ and $Y$ under the null to

$$M \sim I\{A(0) < A(1)\} \left( \text{expit}\{2A - 1 + C_2\} \right)$$
$$\quad + I\{A(0) \geq A(1)\} \left( \text{expit}\{5A - 1 + C_2\} \right)$$
$$Y \sim N \left( \beta M + 2U + 2\sqrt{|C_1|} + \sin(C_4), 1 \right),$$

and we change the equations for $M$ and $Y$ under the alternative to

$$M \sim I\{A(0) < A(1)\}\left(\text{expit}\{2.38A - 1 + C_2\}\right) + I\{A(0) \geq A(1)\}\left(\text{expit}\{5A - 1 + C_2\}\right)$$
$$Y \sim N\left(\beta M + U + 2\sqrt{|C_1|} + \sin(C_4), 1\right).$$

Since there are "defiers" for whom $A(0) = 1$ but $A(1) = 0$, the IV model is invalid.

## C.5 BACKDOOR MODELS WITH DIFFERENT ADJUSTMENT SETS

Finally, we present the data-generating processes for the simulation study combining three backdoor models with different adjustment sets, the results of which are shown in Figure 4 and discussed in Section 4. We simulate data as follows:

$$U \sim \text{Unif}(-2, 2)$$
$$C_i \sim \text{Unif}(-2, 2), \text{ for } i = 1, 2, 3, 4$$
$$A \sim \text{Bern}\left(\text{expit}\{C_1 + C_2\}\right)$$
$$Y \sim N\left(\beta A + 4C_2 + C_3 + U, 1\right).$$

# D   ADDITIONAL SIMULATION RESULTS

Figures 10, 11, and 12 display the size and power of the test for the case of $K = 2$ when combining the backdoor and front-door, backdoor and IV, and front-door and IV models, respectively.

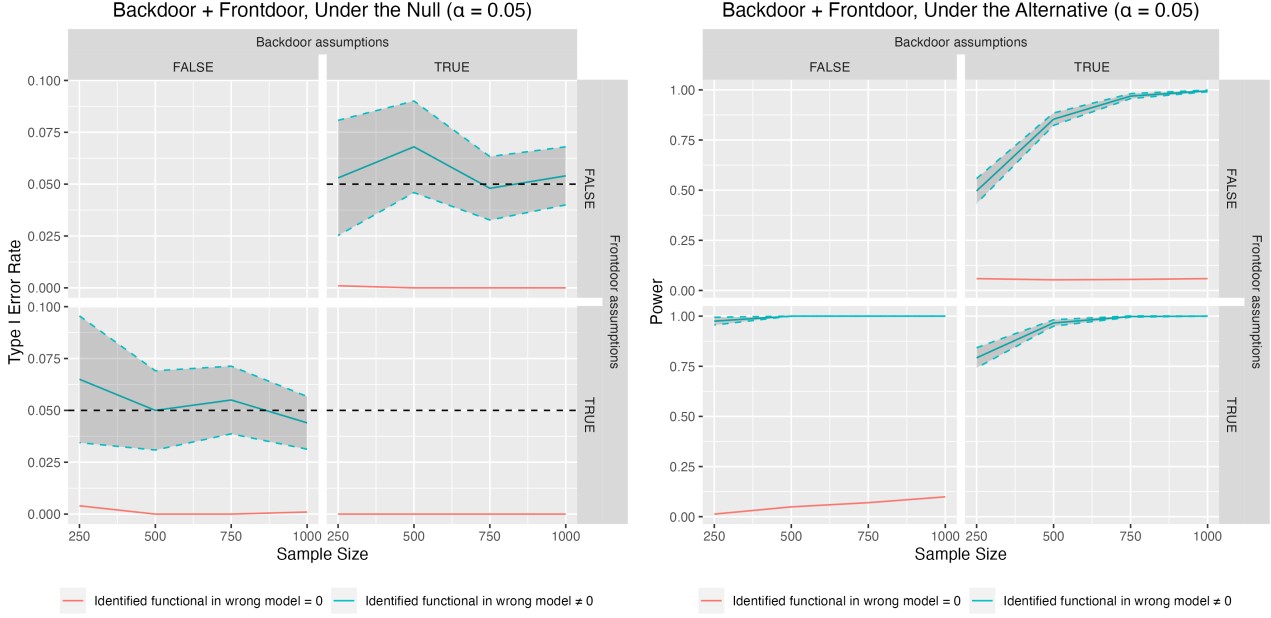

Figure 10: Size (left) and power (right) of the test when combining the backdoor model $\mathcal{M}_1$ and front-door model $\mathcal{M}_2$ when at least one of the models holds.

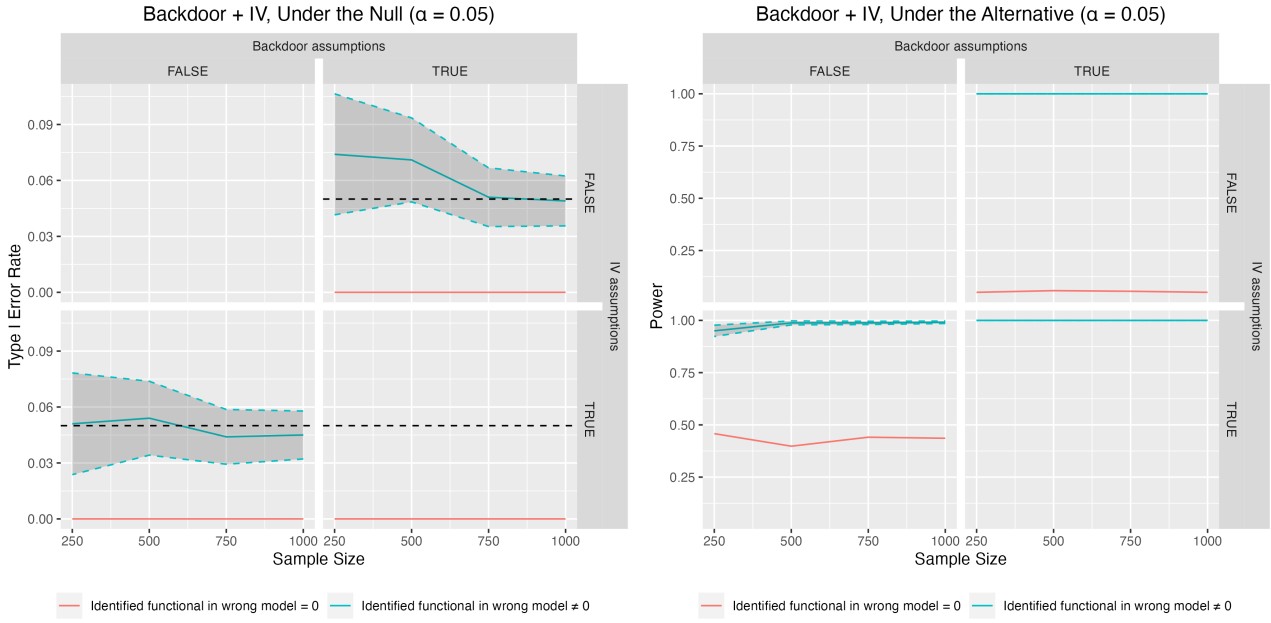

Figure 11: Size (left) and power (right) of the test when combining the backdoor model $\mathcal{M}_1$ and IV model $\mathcal{M}_3$ when at least one of the models holds.

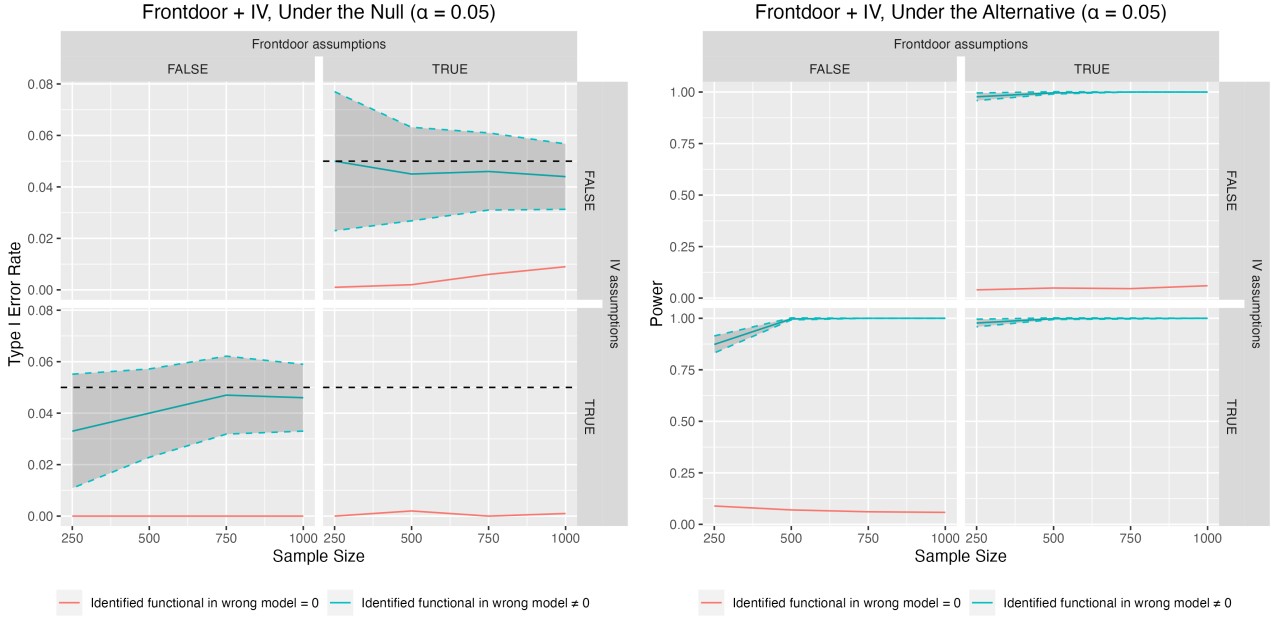

Figure 12: Size (left) and power (right) of the test when combining the front-door model $\mathcal{M}_2$ and IV model $\mathcal{M}_3$ when at least one of the models holds.