# OpenReview forum: "Statistical and Causal Robustness for Causal Null Hypothesis Tests"
_auai.org/UAI/2024/Conference — UAI 2024 oral_

### Official Review · Reviewer_557N · 2024-03-04

**Q2-1 Originality-Novelty:** 2
**Q2-2 Correctness-Technical Quality:** 3
**Q2-5 Clarity Of Writing:** 4

**Q1 Summary And Contributions:**

This paper proposes a method to combine multiple asymptotically linear causal effect estimators to jointly test for the existence of an effect among a set of plausible models. The main idea is to leverage joint asymptotic normality of asymptotically linear estimators.

**Q2-3 Extent To Which Claims Are Supported By Evidence:**

3: Good: the main claims are supported by convincing evidence (in the form of adequate experimental evaluation, proofs, (pseudo-)code, references, assumptions).

**Q2-4 Reproducibility:**

3: Good: key resources (e.g. proofs, code, data) are available and key details (e.g. proofs, experimental setup) are sufficiently well-described for competent researchers to confidently reproduce the main results.

**Q3 Main Strengths:**

I think the authors address a highly significant task of practical causal inference by incorporating model uncertainty. The idea is clearly presented, and the paper is generally well-written.

**Q4 Main Weakness:**

I'm not sure about the (technical) novelty of the paper; the main result seems to be a relatively straightforward application of clt. Further, the usefulness of the method relies on faithfulness to avoid bad cases.

**Q5 Detailed Comments To The Authors:**

At least to me, it would be interesting to further investigate the ('faithfulness') restriction to non-zero targets across invalid models. The implications and importance of this assumption are not fully answered/clear to me.

It would have been interesting to see how well the method performs in simulations compared to multiple testing approaches.

What happens if the causal ordering of A and Y is misspecified, e.g., there is an actual causal effect from Y to A and no effect from A to Y? Is there a way to incorporate this uncertainty over the causal ordering in the framework?

**Q9 Complying With Reviewing Instructions:**

Yes

---

> ### Author Rebuttal · Authors · 2024-04-04
>
> Thank you for your comments! We address them below.
>
> - _At least to me, it would be interesting to further investigate the ('faithfulness') restriction to non-zero targets across invalid models. The implications and importance of this assumption are not fully answered/clear to me_
>
> We do address this question in further detail in the text as well as through simulations, though some of this discussion has been moved to the Supplement in the interest of space. In Appendix B we provide an example of a distribution that violates faithfulness and its corresponding impact on the proposed test statistic. In Figure 2 of the main text, we demonstrate empirically how  violations of faithfulness affect the proposed test. More examples of simulations with unfaithful distributions are in Appendix D.
>
> - _It would have been interesting to see how well the method performs in simulations compared to multiple testing approaches_
>
> Thank your for the suggestion. We agree that evaluation against a baseline would strengthen the paper. However, finding publicly available, real-world data with an accompanying randomized controlled trial (RCT) that can be used as ground-truth (most datasets with ground-truth in causal inference turn out to be synthetic or semi-synthetic) is difficult. A perfect comparison with prior evidence factors work is also challenging as the evidence factors literature typically considers a slightly different class of statistical and causal assumptions. We are, however, currently looking for a dataset that has been used in previous evidence factors analyses to compare our method with prior work. We think we have found one that is suitable---we hope to post the results of this analysis before the discussion period ends, and certainly will in a revision if the paper is accepted.
>
> - _What happens if the causal ordering of A and Y is misspecified, e.g., there is an actual causal effect from Y to A and no effect from A to Y? Is there a way to incorporate this uncertainty over the causal ordering in the framework?_
>
> If the researcher is uncertain about whether $A$ precedes $Y$ or vice versa, in the candidate model where $Y$ precedes $A$, the causal effect of $A$ on $Y$ is identified with 0, so the researcher would never detect a non-zero effect. This uncertainty can be incorporated into our test (see the discussion that follows Theorem 1 regarding the case $\phi_{k,P} = 0$), however, it does not seem to be a very interesting case. This is also true of other methods (e.g., [Jaber et al, 2023](https://openreview.net/pdf?id=OxHn1Yz_Kl3)) that try to estimate causal effects from a Markov equivalence class of causal DAGs where one or more causal DAGs in the equivalence class have the outcome preceding the treatment --- the null effect cannot be ruled out in such cases.

---

### Official Review · Reviewer_a6L6 · 2024-03-13

**Q2-1 Originality-Novelty:** 3
**Q2-2 Correctness-Technical Quality:** 3
**Q2-5 Clarity Of Writing:** 3

**Q10 Ethical Concerns:**

No.

**Q1 Summary And Contributions:**

In this work, the authors develop a method for statistical significance testing of causal effects under the hypothesis that at least one 1 of K prespecified causal models is correct, thereby providing an approach that is robust to model misspecification.

**Q2-3 Extent To Which Claims Are Supported By Evidence:**

3: Good: the main claims are supported by convincing evidence (in the form of adequate experimental evaluation, proofs, (pseudo-)code, references, assumptions).

**Q2-4 Reproducibility:**

4: Excellent: key resources (e.g. proofs, code, data) are available and key details (e.g. proof sketches, experimental setup) are comprehensively described for competent researchers to confidently and easily reproduce the main results.

**Q3 Main Strengths:**

The paper is well-written, the contribution is relevant, and the theory is presented in a concise way.

**Q4 Main Weakness:**

I do not have major complaints, only minor ones that will be discussed below.

**Q5 Detailed Comments To The Authors:**

•	In my view, it would be important to make the implicit trade-off that is happening in your work more apparent. On the one hand, you are relaxing the causal assumption that one causal model is correct to at least one of $K$ models is correct. On the other hand, you need to assume now that for each $k$, it is possible to construct an asymptotically linear estimator, so you now have $K$ statistical assumptions instead of just one in the model with the strong causal assumption. I think that this trade-off is a good one since many estimators are known to have this property, and it can be checked. It should be mentioned more explicitly nevertheless.

•	P.4, the notation $\mathbb P_n$ and the notation $P\phi$ for moments is somewhat non-standard, why not stick to the usual $m$ for mean and $\mathbb E[X^n]$ for moments.

•	It would be helpful to include some plots as in Figure 2 for smaller sample sizes.

•	Do you have any thoughts or heuristics on the trade-off that happens when including more potential models? On the one hand, it seems like this would make it more likely to include the correct model, on the other it raises the probability of including more than one correct model. Any ideas on how to balance this?

**Q9 Complying With Reviewing Instructions:**

Yes

---

> ### Author Rebuttal · Authors · 2024-04-04
>
> Thank you for your positive assessment of our paper and thoughtful comments! We address these comments below.
>
> - _It would be important to make the implicit trade-off that is happening in your work more apparent... I think that this trade-off is a good one since many estimators are known to have this property, and it can be checked. It should be mentioned more explicitly nevertheless._
>
> This is a great point and we agree. We will add a couple sentences in the introduction to discuss this trade-off of causal assumptions for statistical assumptions on the estimators. We will also add a sentence or two to highlight this again in Section 3 where the  discussion of statistical conditions first appears.
>
> - _On notation:_
>
> Thank you for the suggestion, we will change our notation to ones that readers of UAI are more familiar with (this is an easy change since the notation appears in only a few places). The notation we currently use was established in some seminal textbooks on semiparametric statistics (see, e.g., Chapter 25 of van der Vaart's Asymptotic Statistics or Bickel, Klassen, Ritov, and Wellner's Efficient and Adaptive Estimation for Semiparametric Models); this was our original motivation for the notation used.
>
> - _It would be helpful to include some plots as in Figure 2 for smaller sample sizes._
>
> We have repeated the experiment for Figure 2 with sample size 100. The new results follow similar trends and will be included in our revision of the paper.
>
> - _Do you have any thoughts or heuristics on the trade-off that happens when including more potential models? On the one hand, it seems like this would make it more likely to include the correct model, on the other it raises the probability of including more than one correct model. Any ideas on how to balance this?_
>
> This is a great question. We agree that including more models makes it more likely that a correct model is included, but including more models can also reduce the power of the test if some of the identified functionals in the incorrect models are close to zero. A precise study of the power of the method as a function of the number of causal models would be an interesting line of future research. Our general recommendation is to use a few causal models that have substantive differences, rather than many models of the same type that have little meaningful difference. This is why we have focused in the paper on comparing substantively different models like backdoor, front-door, and IV. An example of an approach we would be wary of would be simply enumerating all possible subsets of covariates as possible backdoor adjustment sets, which leads to $2^d$ possible candidate models if there are $d$ candidate adjustment covariates. In addition to possible computational problems, we expect this approach could have very low power under the alternative, even if one of the candidate sets turns out to be correct. Instead, we would recommend that researchers consider sets of adjustment covariates that come from scientifically-grounded candidate causal models.  We would be happy to add discussion of these points to the paper with the additional page granted if the paper is accepted.

---

### Official Review · Reviewer_gCdB · 2024-03-20

**Q2-1 Originality-Novelty:** 2
**Q2-2 Correctness-Technical Quality:** 3
**Q2-5 Clarity Of Writing:** 3

**Q1 Summary And Contributions:**

The authors propose a method for improving the robustness of causal null hypothesis tests by performing a joint test on multiple causal models. They show that, if one of the considered causal models is true, then the proposed test has asymptotically valid type I error rate, while its power goes to one under fixed alternative hypotheses. Moreover, they show in their experiments that the test is generally conservative if two or more of the causal models are true.

**Q2-3 Extent To Which Claims Are Supported By Evidence:**

3: Good: the main claims are supported by convincing evidence (in the form of adequate experimental evaluation, proofs, (pseudo-)code, references, assumptions).

**Q2-4 Reproducibility:**

3: Good: key resources (e.g. proofs, code, data) are available and key details (e.g. proofs, experimental setup) are sufficiently well-described for competent researchers to confidently reproduce the main results.

**Q3 Main Strengths:**

The paper is very well organized and clearly written. The technical quality of the work seems high and the claims are adequately supported by experimental evidence.

**Q4 Main Weakness:**

The only weakness I perceive is that the methodology proposed is perhaps a rather straightforward extension of individual (causal) hypothesis testing to joint (causal) hypothesis testing.

**Q5 Detailed Comments To The Authors:**

- How come you only include one numerical experiment for Section 4.2 in Figure 3? It seems like enough space could have been made to add extra plots, or at least a short description of the results for the other two experiments. For example, the power analysis in Figure 3 does not give a lot of information, so maybe it would have been more interesting to look at the type I error rate for the other two adjustments sets.
- Page 4, before Section 3.1: It would have been useful to first introduce the notation $P{\phi_{k, P}}$ ($P{\phi^2_{k, P}}$), which seems overloaded, as $P$ appears twice. Am I right that these quantities are the same $\mathbb{E}[\phi_{k, P}]$ ($\mathbb{E}[\phi^2_{k, P}]$), in which case was it necessary to introduce this notation at all? Furthermore, it would have been useful to have a small section on the notation used, since the reader might not be familiar with all the terminology.
- Page 5, Theorem 2 and last line: $\sum_{j=1}^K \psi_{k,P}$ should be $\sum_{k=1}^K \psi_{k,P}$.

**Q9 Complying With Reviewing Instructions:**

Yes

---

> ### Author Rebuttal · Authors · 2024-04-04
>
> Thank you for your positive evaluation of the paper. We address your comments below.
>
> - _How come you only include one numerical experiment for Section 4.2 in Figure 3?_
>
> Correct us if we are misinterpreting your question: Is the question asking why we chose to include only one of a few different numerical studies we conducted for this particular setting? If so, we think there might be a slight misunderstanding with the language we used in the paragraph describing the numerical study. We say:  ``We use our proposed test with three AIPW estimators with the three different adjustment sets\dots Figure 3 displays the results of the second numerical study." The latter sentence is just referring to the fact that Figure 3 shows the results of the simulation study in Section 4.2, which is the second of two simultion studies in the paper (the first being in Section 4.1), as opposed to a particular version of the numerical study considered in Section 4.2.
>
> We have also extended our numerical studies for Figure 2 to include lower sample sizes (as suggested by another reviewer) --- the new results follow similar trends and will be included in our revision of the paper.
>
> - _Re: undefined notation_
>
> Thank you for pointing this out, your interpretation of the notation is correct. We will also change our notation to ones that readers of UAI are more familiar with (this is an easy change since the notation appears in only a few places). The notation we currently use was established in some seminal textbooks on semiparametric statistics (see, e.g., Chapter 25 of van der Vaart's  _Asymptotic Statistics_ or Bickel, Klassen, Ritov, and Wellner's _Efficient and Adaptive Estimation for Semiparametric Models_); this was our original motivation for the notation used.
>
> - _Re: typo_
>
> Thank you for catching the typo! We have fixed it.

---

### Official Review · Reviewer_gsuN · 2024-03-24

**Q2-1 Originality-Novelty:** 3
**Q2-2 Correctness-Technical Quality:** 4
**Q2-5 Clarity Of Writing:** 4

**Q1 Summary And Contributions:**

The paper introduces a statistical test for testing the null hypothesis that no causal effect is present between the treatment and the outcome. The test is robust as it works on a set of potential causal models, requiring only one of them to be correctly specified. The distribution of the test statistic under null is derived analytically under the assumption that semi-parametric linear models are used for estimating the causal effect.

**Q2-3 Extent To Which Claims Are Supported By Evidence:**

4: Excellent: all claims are supported by very convincing evidence (in the form of comprehensive experimental evaluation, rigorous mathematical proofs, detailed (pseudo-)code, precise references, well-motivated and realistic assumptions) and the authors deliver what they promise.

**Q2-4 Reproducibility:**

4: Excellent: key resources (e.g. proofs, code, data) are available and key details (e.g. proof sketches, experimental setup) are comprehensively described for competent researchers to confidently and easily reproduce the main results.

**Q3 Main Strengths:**

Model selection is a challenging problem in causal inference and researchers typically rely on domain knowledge for it. For the problem statement of testing whether a causal effect is present, the proposed approach avoids the model selection step by working on a set of potential causal models. The test is straightforward to implement and use in practice, and p-values can be computed analytically.

The paper is well written, and the example helps in understanding both the problem statement and the proposed solution.

**Q4 Main Weakness:**

In my opinion, researchers are typically not only interested in testing for the presence of a causal effect but also in estimating them. As the proposed method can only test for the presence of a causal effect, the potential application of the method in practice seems limited. Additionally, the edge cases of the test, especially that no more than one correctly specified model can be present in the potential models set, could make it more challenging to use in practice.

**Q5 Detailed Comments To The Authors:**

I am not aware of studies in which the main research question is only testing for the presence/absence of a causal effect. I think the method's practical applicability would be more convincing if some references to such studies could be added.

**Q9 Complying With Reviewing Instructions:**

Yes

---

> ### Author Rebuttal · Authors · 2024-04-04
>
> Thank you for your comments! We address your main comments below.
>
> - _In my opinion, researchers are typically not only interested in testing for the presence of a causal effect but also in estimating them. As the proposed method can only test for the presence of a causal effect, the potential application of the method in practice seems limited.... I am not aware of studies in which the main research question is only testing for the presence/absence of a causal effect. I think the method's practical applicability would be more convincing if some references to such studies could be added._
>
> We agree that effect estimates are important in many settings, but testing for a null effect can also be an important goal. Besides the evidence factors work cited in our paper where the focus is on causal null hypothesis testing, there are also other applied fields where researchers have focused on this: e.g., [Swanson et al, 2018](https://www.ncbi.nlm.nih.gov/pmc/articles/PMC6061140/) in epidemiology, [Eggers et al, 2023](https://onlinelibrary.wiley.com/doi/full/10.1111/ajps.12818) in political science, and [Angrist and Kuersteiner, 2011](https://www.jstor.org/stable/23016073) in economics. Testing for a causal null is also often considered when the estimated local effect on a particular subpopulation is obtained by particular designs, such as, instrumental variables or regression discontinuity; our proposed framework can be applied here as well.
>
> In addition, it may be possible to use the proposed methods to construct confidence intervals for the causal effect that are valid as long as at least one of the causal models are valid by first developing analogous tests of the null hypothesis that $\beta = c$, then inverting the resulting test. We can add a brief discussion of the importance of effect estimates and this extension to the discussion.
>
>
> - _Additionally, the edge cases of the test, especially that no more than one correctly specified model can be present in the potential models set, could make it more challenging to use in practice._
>
> We would like to push back on this comment---the test can still be used when more than one model is correctly specified; the corresponding inferences are still correct though they may be conservative. We demonstrate this in our simulations in Section 4: some of the panels in Figure 2 emphasize how the test operates under conditions where more than one model e.g., backdoor and IV, are correct. We also discuss some of the theoretical difficulties one might encounter when trying to design a method to obtain exact error rates with multiple correct models in Section 6, and pose some ideas for future work on this topic.

---

### Official Review · Reviewer_pv4V · 2024-03-31

**Q2-1 Originality-Novelty:** 3
**Q2-2 Correctness-Technical Quality:** 3
**Q2-5 Clarity Of Writing:** 3

**Q1 Summary And Contributions:**

This paper considers the problem of testing of a causal effect (using observational data) under model misspecification. Specifically, consider K models; each of which identify the causal effect using a different estimand and an asymptotically linear influence-function based estimator. This paper proposes a test for a relaxed (or implied) null hypothesis of testing whether at least one of the estimands is 0. Note that this corresponds to the original null hypothesis only if at least one of the models is the correct one. Finally, the test is studied under synthetic data and real-world data.

**Q2-3 Extent To Which Claims Are Supported By Evidence:**

3: Good: the main claims are supported by convincing evidence (in the form of adequate experimental evaluation, proofs, (pseudo-)code, references, assumptions).

**Q2-4 Reproducibility:**

3: Good: key resources (e.g. proofs, code, data) are available and key details (e.g. proofs, experimental setup) are sufficiently well-described for competent researchers to confidently reproduce the main results.

**Q3 Main Strengths:**

The main strength of the paper lies in the introduction of the formulation of testing causal model misspecification. While implicitly this was considered in earlier works on evidence factors, perhaps an end-to-end test that also focuses on the statistical estimation aspect was not developed. The proposed method also has multiple advantages that improve upon the evidence factors literature. The paper is clearly written and the problem is well-motivated. The proofs are correct to my understanding.

**Q4 Main Weakness:**

I don't have specific main weaknesses to point out. So, I will add some major comments and leave more minor comments for the next section.
1) Aren't Theorem 1 and 2 true only if at least one of the causal models under consideration is correctly specified? I think this is important to specify explicitly.
2) Is it possible to do more validation on other real-world datasets? The reason is that the current dataset under study doesn't seem to have any discussion about a particular baseline to compare against. Can these results be compared perhaps with the evidence factors tests?

**Q5 Detailed Comments To The Authors:**

1. I was confused by the conclusion of Section 4.2 and the claimed advantage iii in Section 1. The former claims that unless bias influences all candidate models, the test is valid whereas the latter makes no such reference to this clause.
2. For the IV setting in the numerical study in 4.1, violating the assumptions mentions "include unmeasured confounding between Z and A". This alone doesn't violate the IV assumption though.

**Q9 Complying With Reviewing Instructions:**

Yes

---

> ### Author Rebuttal · Authors · 2024-04-04
>
> Thank you for your positive assessment of the paper and thoughtful questions! Please find below a detailed response to your questions.
>
> - _Aren't theorems 1 and 2 true only if at least one of the causal models is correctly specified?_
>
> Thanks for the clarifying question. Theorems 1 and 2 establish  properties of our test of the null hypothesis $H_0$ that the product of observed data parameters $\prod_{k=1}^K \psi_{k, P}$ is equal to zero from a purely statistical viewpoint. This statistical null hypothesis test  has an added interpretation as a causal null hypothesis test when at least one of the $K$ causal models is correct so that some $\psi_{k, P}=\beta$. We chose to emphasize this separation in the phrasing of the theorems, because it is helpful to draw a distinction between the statistical issues associated with estimation and the identification problem of the causal parameter -- ``once a causal parameter is identified, causal inference reduces to a statistics problem'' is a common adage. We will make this clearer by adding the explicit causal interpretations of these theorems under the appropriate causal conditions to the discussion following the theorems.
>
> - _Is it possible to do more validation on other real-world datasets with established baselines?_
>
> We agree that evaluation with another real-world dataset with a baseline would strengthen the paper. However, finding publicly available, real-world data with an accompanying randomized controlled trial (RCT) that can be used as ground-truth (most datasets with ground-truth in causal inference turn out to be synthetic or semi-synthetic) is difficult. A perfect comparison with prior evidence factors work is also challenging as the evidence factors literature typically considers a slightly different class of statistical and causal assumptions. We are, however, currently looking for a dataset that has been used in previous evidence factors analyses to compare our method with prior work. We think we have found one that is suitable---we hope to post the results of this analysis before the discussion period ends, and certainly will in a revision if the paper is accepted.
>
> - _Re: clause in Section 1_
>
> Yes (iii) in Section 1 also has a similar clause -- at least one causal model must be correctly specified. We will edit the phrasing to make this more clear.
>
> - _Re: the description of the IV setting in the numerical study
>
> Thank you for pointing out the ambiguity. We will fix the phrasing to say that we sometimes include a $U \rightarrow Z$ edge in the data generating process to violate the IV assumptions. This edge in conjunction with the $A \leftarrow U \rightarrow Y$ edge implies that all of $A, Z, Y$ share a common unmeasured confounder $U$, which violates the IV assumptions.

---

### Meta-Review · Area_Chair_qWmf · 2024-04-16

Here we have a quite unusual agreement of positive reviews and scores. Reviewers acknowledge relevance, technical quality and presentation.